# Quantum beats of exciton-polarons in CsPbI$_3$ perovskite nanocrystals

Artur V. Trifonov [1,2] ✉, Mikhail O. Nestoklon [1], M. Alex Hollberg [1], Stefan Grisard [1], Dennis Kudlacik[1], Elena V. Kolobkova [3,4], Maria S. Kuznetsova [2], Serguei V. Goupalov [5], Jan M. Kaspari [6], Doris E. Reiter [6], Dmitri R. Yakovlev [1], Manfred Bayer [1,7] & Ilya A. Akimov [1] ✉

The optical response of semiconductors is governed by coupled electronic and vibrational excitations. In lead-halide perovskite nanocrystals, strong exciton–phonon interaction forms a ladder of exciton-polaron states accessible by femtosecond laser pulses. We demonstrate a fully coherent regime of exciton-polaron dynamics with long optical coherence times ($T_2 \approx 300$ ps) in CsPbI$_3$ nanocrystals embedded in glass. Using transient two-pulse photon echo at a temperature of 2 K, we observe quantum beats between exciton-polaron states, with decay determined by optical phonon lifetimes of 5-15 ps. Within a four-level model, we directly quantify the exciton–phonon coupling strength through Huang–Rhys factors of $0.05 - 0.12$ and $0.02 - 0.04$ for low-energy optical phonons with energies of 3.2 and 5.1 meV, respectively. The pronounced size dependence of both coupling strengths and phonon lifetimes offers a route to tune the optical transitions between exciton-polaron states and tailor the coherent optical dynamics in perovskite semiconductors for solid-state quantum technologies.

Lead halide perovskite semiconductors attract close attention due to their intriguing optical and electrical properties with huge potential for photovoltaic and light emitting applications[1,2]. These materials provide a unique playground for polaron physics in the solid state governed by the interaction of electronic excitations with the phonons of the crystal lattice[3]. In particular, several studies indicate that low-energy optical phonons in halide perovskites have an important influence on the conductivity, while acoustic phonons have a minor effect[4,5]. This is in contrast to conventional semiconductors such as GaAs, where the interaction of electrons with acoustic phonons is dominant at cryogenic temperatures[6,7]. Furthermore, the carrier mobility is determined not by a few specific optical phonons, but by a large multitude of modes due to the complex phonon spectrum[8].

Perovskite nanocrystals (NC) belong to the class of quantum emitters with discrete energy level spectrum, large spectral tunability, and high quantum yield, which makes them attractive as single photon sources[9,10]. At cryogenic temperatures, the elementary optical excitation in NCs is an exciton (electron-hole pair) which possesses long optical and spin coherence times up to hundreds of ps, comparable to its radiative lifetime[11–13]. Long-lived coherent excitons are often considered as qubits and are appealing for possible applications in quantum communication[14]. A distinctive feature of perovskite nanocrystals is the existence of low-energy optical phonons with a strong coupling to excitons. The strong exciton-phonon interaction leads to formation of exciton-polarons which modifies the energy level spectrum. Exciton-polarons are defined here as confined excitons dressed

[1]Experimentelle Physik 2, Technische Universität Dortmund, Dortmund, Germany. [2]Spin Optics Laboratory, St. Petersburg State University, Peterhof, St. Petersburg, Russia. [3]ITMO University, St. Petersburg, Russia. [4]St. Petersburg State Institute of Technology, St. Petersburg, Russia. [5]Department of Physics, Jackson State University, Jackson, Mississippi, USA. [6]Condensed Matter Theory, Department of Physics, Technische Universität Dortmund, Dortmund, Germany. [7]Research Center FEMS, Technische Universität Dortmund, Dortmund, Germany. ✉e-mail: artur.trifonov@tu-dortmund.de; ilja.akimov@tu-dortmund.de

and renormalized by lattice distortion. This definition is particularly appropriate for quantum dots, where electronic excitations are confined in all three dimensions and the difference between localized and delocalized states is not as drastic as in quasi-1D, -2D, or bulk crystals[15]. In particular, the photoluminescence spectra of single NCs show pronounced optical phonon replicas which confirm the substantial exciton-phonon coupling with Huang-Rhys factors in the range from 0.05 to 0.6 in lead halide perovskite NCs[16–21]. Recently, electron-phonon interaction dominated by optical phonons with comparable Huang-Rhys factor of about 0.4 was demonstrated in CsPbBr$_3$ NCs[22] and other inorganic lead halide perovskite NCs[23] using Raman spectroscopy. Compared to InGaAs quantum dots[24,25], these values are significantly larger and close to those reported in II-VI nanostructures[26].

The exciton-polaron dynamics can be directly probed by time-resolved studies. It is well established that scattering on incoherent phonons with a thermal distribution causes loss of exciton coherence[27,28]. Yet, less is known about the quantum dynamics of exciton-polarons. Transient pump-probe spectroscopy provides access to coherent superpositions of exciton states originating from both the exciton fine structure and excited exciton-polaron states. These studies have revealed the optical excitation of vibrational wave packets and inter-exciton coherent dynamics modulated by coherent and thermal phonons[29–33]. Notably, the simultaneous observation of exciton and phonon coherences reported in ref. 31 indicates the possibility of fully coherent exciton-polaron dynamics on timescales exceeding the phonon oscillation period. At the same time, recent studies showed that correlated lattice fluctuations can support long-lived inter-exciton coherence without preservation of optical coherence[33,34]. Therefore, direct probing of quantum dynamics of the individual states and their correlations in the optical domain is essential.

Recently, transient four-wave mixing and two-dimensional Fourier spectroscopy also revealed the importance of coherent exciton-phonon coupling for the ground and excited (exciton and biexciton) states in lead halide perovskites[33,35–38]. However, most of these studies were limited to the case of short optical coherence of excitons (≤100 fs) and therefore reported on the coherent evolution of either the exciton or the ground electronic state modulated by the frequencies of phonons with much slower decoherence. In other words, the lack of optical coherence restricted the observations to the phonon dynamics, while the genuine coherent evolution of the exciton-polarons has remained inaccessible. This scenario was also documented for bulk semiconductors, nanostructures and organic molecules[39–43]. However, the opposite quantum dynamics limit, where pronounced quantum beats of exciton-polarons are expected, has not yet been uncovered. It should be noted that two-dimensional Fourier spectroscopy has also been used to resolve the free polaron dynamics at room temperature in the adiabatic strong-coupling regime[33,36,44,45], a scenario that is fundamentally different from the coherent dynamics of confined exciton-polarons.

Another general question arises concerning the limitations of coherent control under resonant excitation with femtosecond pulses. Time-resolved studies have reported rather short coherence times, up to 25 ps, in lead halide perovskites even at cryogenic temperatures[35,46], in contrast with photoluminescence data of single NCs[11–13]. A possible reason is that most coherent spectroscopy studies have been performed on closely packed NC ensembles, which are known to exhibit significantly different coherent dynamics compared with single NCs[47–49]. Therefore, it is also important to study diluted NCs embedded in a glass matrix with a NC density in the range of $10^{15}$ cm$^{-3}$[50,51], where the NCs can be treated as isolated and non-interacting.

In this work we establish a fully coherent dynamics of exciton-polarons in NCs with an exceptionally long optical coherence time of the zero-phonon transition ($T_{2,X} \approx 300$ ps) on the one hand and a discrete spectrum of spectrally narrow optical phonon modes on the other hand. The regime manifests in exciton-polaron quantum beats, which are detected by transient two-pulse photon echoes in an ensemble of CsPbI$_3$ nanocrystals embedded in a glass matrix at low temperature, $T = 2$ K. The oscillatory signal is determined by optical phonons with low frequencies corresponding to 3.2 and 5.1 meV energy, and can be described by the solution of the Lindblad equations for a set of four-level systems that include a phonon ladder in the ground and optically excited states. Different sizes, with diameters ranging from 12 to 15 nm, were selectively addressed within the same ensemble of nanocrystals by varying the laser photon energy. From the relative amplitude of oscillations we directly evaluate the strength of the exciton-phonon coupling with Huang-Rhys factors of 0.05 and 0.02 for optical phonons with 3.2 meV and 5.1 meV energy, respectively. Correspondingly, the electron-phonon coupling strength doubles reaching values of 0.12 and 0.04 as the NC diameter is reduced from 15 to 12 nm. The theoretical analysis of the size dependence shows that both the deformation potential and the Fröhlich mechanism of electron-phonon interaction qualitatively agree with the experimental data in the regime of weak exciton confinement. Our study demonstrates that the dynamics of a pure quantum state following resonant optical excitation requires consideration of exciton-polaron effects, whose strength and coherence can be controlled by composition and size of the NCs. This opens up new possibilities for coherent control of exciton-polaron dynamics in lead-halide perovskite NCs.

## Results

### Photon echo and long coherence of zero-phonon exciton

The ensemble of CsPbI$_3$ NCs is synthesized in a fluorophosphate glass matrix by rapid cooling of a glass melt enriched with the materials needed for the perovskite crystallization. The details of synthesis are given in ref. 50. The NC size is about 12 – 15 nm (the details on the properties of the studied nanocrystals are provided in Supplementary Notes 1 and 2). The low-temperature photoluminescence (PL) spectrum is dominated by a 60 meV broad band centered around 1.75 eV, as shown in Fig. 1a. The broadening originates from fluctuations of the NC size, which we estimate to be in the order of 20%. A single NC is schematically illustrated in the inset of Fig. 1b. Note that the cubic crystal structure is shown for illustrative purposes as the actual crystallographic phase depends on the growth conditions and the size of the nanocrystals[52–55].

We perform transient four-wave mixing (FWM) experiments in transmission geometry as shown schematically in Fig. 1b. The NCs are resonantly excited with a sequence of two 120 fs laser-pulses with photon energies $h\nu < 1.76$ eV in the low energy flank of the PL band. The sample temperature is kept at $T = 2$ K. The electric field amplitude of the FWM signal $\mathcal{E}_{FWM}(t)$ is resolved in time using heterodyne detection where the signal field is temporally overlapped with a strong reference pulse with amplitude $\mathcal{E}_{ref}(t - \tau_{ref})$ (see "Methods" and refs. 56–59 for details). Here, time $t = 0$ corresponds to excitation with the first pulse, while $\tau_{ref}$ is the delay of the reference pulse with respect to the first pulse in the excitation sequence. The resulting FWM signal is shown in Fig. 1c, which gives a two-dimensional plot of $\mathcal{E}_{FWM}$ as function of the delay time between the first and second pulses $\tau_{12}$ (vertical axis) and the reference delay time $\tau_{ref}$ (horizontal axis). The FWM signal demonstrates the expected peak centered at time $\tau_{ref} = 2\tau_{12}$, corresponding to emission of a photon echo (PE) from the NC ensemble[60]. This behavior arises from the inhomogeneous broadening of the optical transitions resulting from fluctuations of the NC size as demonstrated for similar halide perovskite NCs[35,46]. Interestingly, during the first tens of ps the PE amplitude shows pronounced high frequency oscillations with increasing $\tau_{12}$ which we will discuss below.

On a longer time scale, the amplitude of the photon echo decays exponentially with increasing delay time $\tau_{12}$ as shown in Fig. 1d. Using

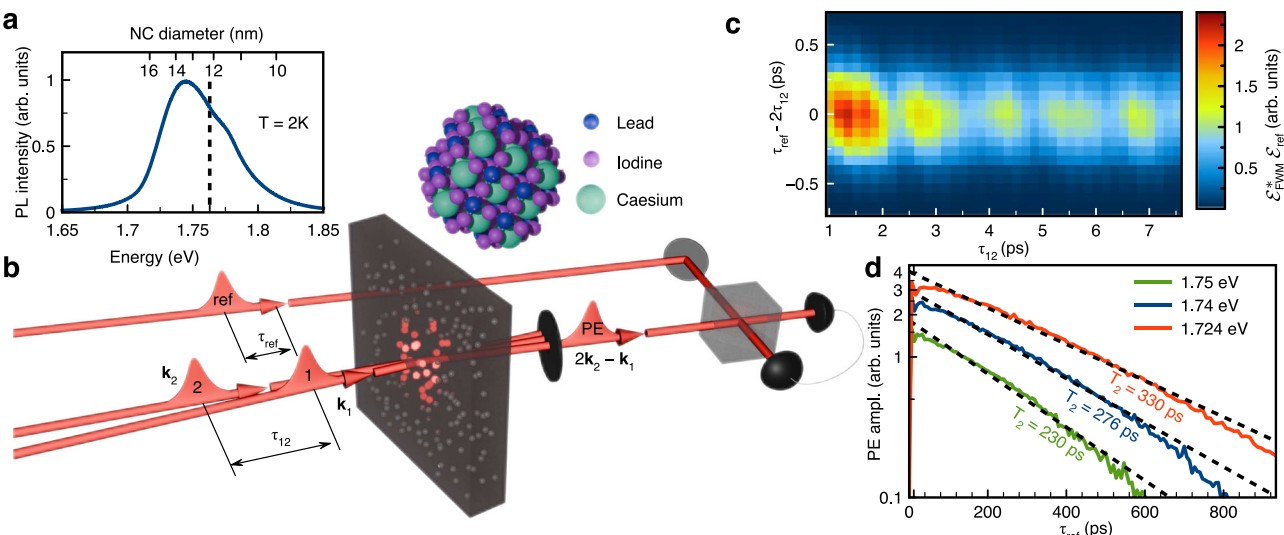

**Fig. 1 | Photon echoes in CsPbI₃ nanocrystals. a** Typical photoluminescence (PL) spectrum for excitation with photon energy of 2.33 eV. Vertical dashed line indicates the highest photon energy used in the four-wave mixing FWM experiment. The scale on top corresponds to the nanocrystal (NC) diameter. **b** Schematic representation of the experimental geometry, where the laser pulses hit the sample with wave vectors $\mathbf{k}_1$, $\mathbf{k}_2$. The photon echo is detected along the direction $2\mathbf{k}_2 - \mathbf{k}_1$ using a heterodyning technique by overlapping it with a reference pulse. Inset shows schematically a CsPbI₃ perovskite nanocrystal. **c** Two-dimensional plot of the FWM electric field amplitude $\mathcal{E}^*_{\mathrm{FWM}}$ as function of delay between the two pulses $\tau_{12}$ and the reference time $\tau_{\mathrm{ref}}$. The photon echo (PE) signal forms at the time $\tau_{\mathrm{ref}} = 2\tau_{12}$. The amplitude of the PE shows oscillations during the initial evolution when $\tau_{12}$ is scanned. The signal is recorded in linearly co-polarized configuration. Photon energy $h\nu = 1.736$ eV. **d** Decay of two-pulse photon echo amplitudes for excitation with different photon energies. The data are measured with ps excitation pulses, which gives better spectral resolution (see also Supplementary Note 3 for more details). Black dashed lines are fits with exponential functions, from which the labeled exciton coherence times $T_2$ are extracted. Temperature $T = 2$ K.

exponential decay fits, shown with the dashed lines, we obtain $T_2 = 150 - 330$ ps for photon energies in the range $1.724 - 1.765$ eV, showing a smooth decrease with increasing $h\nu$ (for details on the spectral dependence of $T_2$ see Supplementary Section 3). The population relaxation time $T_1$, measured in three-pulse experiments, is found to be $T_1 = 600 - 800$ ps, and is comparable to the exciton lifetime measured by the time-resolved PL on a similar sample[61]. Therefore, we conclude that the signal at $\tau_{\mathrm{ref}} > 50$ ps is due to the zero-phonon transition with long-lived optical coherence time $T_2$ and exciton recombination lifetime $T_1$. If the coherent dynamics were governed only by the population decay we would expect the relation $T_2 = 2T_1$ to hold, which is the case for excitons in self-assembled InGaAs quantum dots at $T = 2$ K[62]. Here, we observe a different situation, where elastic scattering processes (pure dephasing) with a decay constant of $T_{\mathrm{p}} = [1/T_2 - 1/(2T_1)]^{-1} = 330$ ps mainly govern the exciton coherence. Nevertheless, to the best of our knowledge, the homogeneous linewidth of the zero-phonon exciton of $\Gamma_2 = 2\hbar/T_2 = 4.8\,\mu$eV demonstrated here has a record low value for perovskite nanocrystals (see Supplementary Table 1 in Supplementary Note 3).

## Coherent exciton-phonon dynamics

As follows from Fig. 1c for short delay times $\tau_{12} \lesssim 10$ ps, oscillations of the PE signal with multiple frequencies are observed. For deeper insight into the origin of the oscillations, we analyze polarization-resolved PE signals. These results are summarized in Fig. 2a–c. We employ two configurations, where the first and second pulses are linearly co- or cross- polarized, while the detection polarization in all cases coincides with that of the first pulse (see "Methods"). The PE amplitudes $A_\parallel$ and $A_\times$ as functions of $\tau_{12}$ in co-($\parallel$) and cross-($\times$) polarized configurations are shown in Fig. 2a. The oscillations can be grouped into two categories: slow oscillations with a period of 1–10 ps and fast oscillations with a period of less than 1 ps.

It is evident that the fast oscillations are in phase and nearly identical for both polarization configurations. We will show below that these oscillations correspond to quantum beats between

exciton-polaron states. By contrast, the slow oscillations are out of phase. The co-polarized $A_\parallel$-signal starts at its maximum value, while the $A_\times$-signal starts from zero. Subsequently, the minimum of the slow oscillations in $\parallel$-configuration around $\tau_{12} = 5$ ps corresponds to a maximum in the ×-configuration. This out-of-phase behavior is related to quantum beats between the orthogonally polarized bright exciton states[46]. In perovskite nanocrystals, the bright exciton fine structure comprises three linearly polarized states[17,63,64], split by the energies $\delta_1$ and $\delta_2$ as shown in Fig. 3a. This energy level scheme is similar to the fine structure splitting in self-assembled quantum dots[65,66] and in II-VI colloidal NCs[67–69]. We do not consider dark states with total angular momentum zero, as they cannot be optically addressed and, owing to the slow energy relaxation at low temperatures, do not contribute to the coherent dynamics. In order to isolate the oscillations due to the fine structure splitting in the data, we model the expected polarization dependence of the PE signal, accounting also for the random orientation of the nanocrystals (for details see Supplementary Note 4). We obtain the following equations for the amplitude of the phonon echo signal in two polarization configurations:

$$
\begin{aligned}
A_\parallel &= \left[ \frac{18}{15} + \sum_i \frac{4}{15} \cos(\delta_i \tau_{12}) e^{-\frac{2\tau_{12}}{t_i^*}} \right] \Psi_0(\tau_{12}), \\
A_\times &= \left[ \frac{6}{15} - \sum_i \frac{2}{15} \cos(\delta_i \tau_{12}) e^{-\frac{2\tau_{12}}{t_i^*}} \right] \Psi_0(\tau_{12}),
\end{aligned}
\tag{1}
$$

where the index $i = 1, 2, 3$ corresponds to the beats between the three optical transitions ($\delta_3 = \delta_1 + \delta_2$), and $t_i^*$ is the dephasing time of the beats caused by the dispersion of the splitting energies $\hbar\delta_i$ in the ensemble. The common factor $\Psi_0(\tau_{12})$ is the same for both polarization configurations and will be discussed below, see also Supplementary Note 4. Following Eqs. (1) it is possible to distinguish between quantum beats related to the exciton fine structure and other polarization insensitive contributions to the PE signal by introducing the

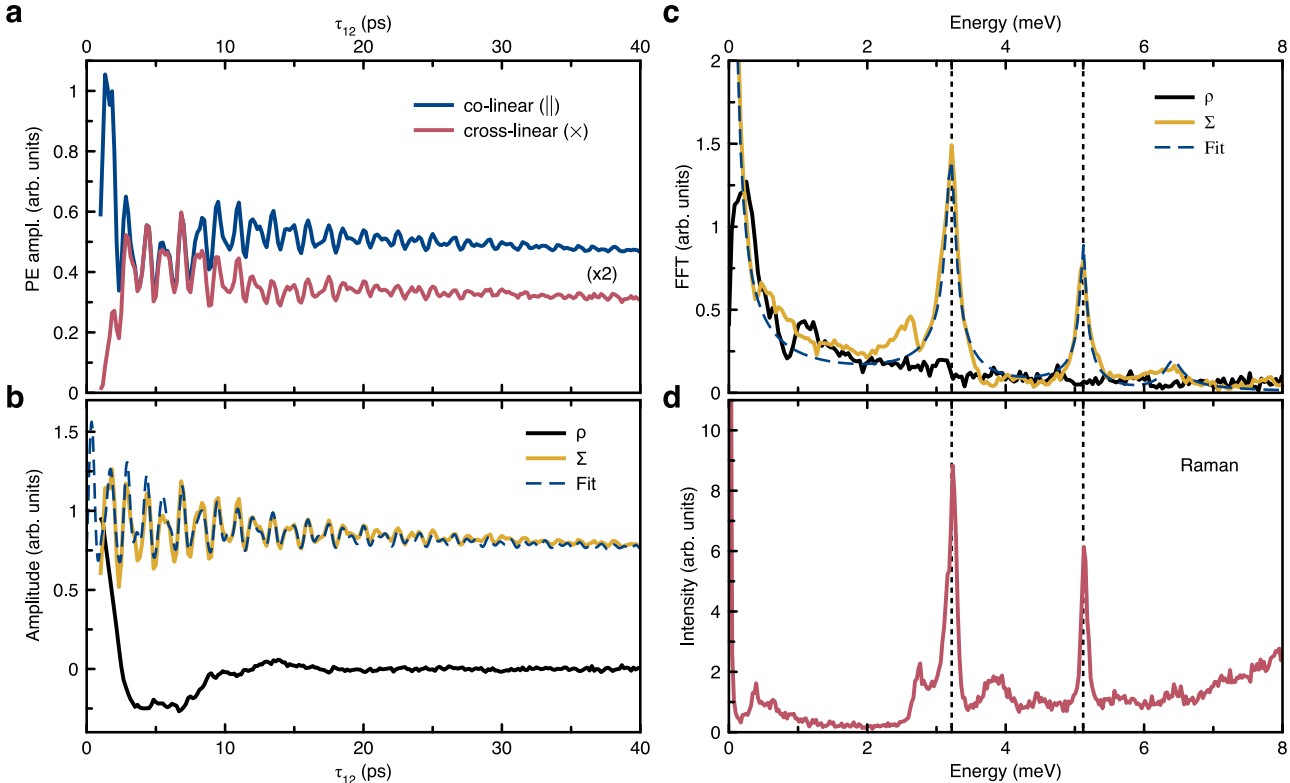

**Fig. 2 | Quantum beats of exciton-polarons. a** Initial range of the PE dynamics in the co-linear (‖) and cross-linear (×) polarization configurations shown with blue and red lines, respectively. Photon energy $h\nu = 1.746$ eV. The amplitude of the × signal is multiplied by two for clarity. **b** Dynamics of the polarizaton-dependent combination $\rho$ (black) and polarization-independent one $\Sigma$ (yellow) as defined by Eqs. (2) and (3), respectively. The dashed blue curve is a fit using the exciton-polaron model with Eq. (5) with the following parameters for two phonon modes:

$\hbar\Omega_1 = 3.2$ meV, $S_{HR1} = 0.097$, $\tau_{ph1} = 5.1$ ps; $\hbar\Omega_2 = 5.1$ meV, $S_{HR2} = 0.032$, $\tau_{ph2} = 10$ ps. **c** Fast Fourier transform (FFT) amplitude spectra of $\rho$ (black), $\Sigma$ (yellow), and the fit curves in panel b. Vertical dashed lines mark the peaks corresponding to optical phonons. **d** Raman spectrum measured at photon energy $h\nu = 1.734$ eV. Vertical dashed lines indicate the positions of peaks corresponding to the optically active optical phonon modes that couple most strongly to the exciton.

polarization contrast

$$\rho = \frac{A_{\parallel} - 3A_{\times}}{A_{\parallel} + 2A_{\times}}, \tag{2}$$

and the polarization sum

$$\Sigma = \frac{A_{\parallel} + 2A_{\times}}{2} = \Psi_0(\tau_{12}). \tag{3}$$

It follows that the temporal evolution of $\rho$ exhibits oscillations at frequencies corresponding to the energy splittings $\delta_i$ of the exciton fine structure and is independent of $\Psi_0(\tau_{12})$. In contrast, the polarization sum $\Sigma$ is given by the intrinsic coherent dynamics $\Psi_0(\tau_{12})$ only.

Figure 2b shows the dynamics of these quantities, calculated from the data in Fig. 2a. In full accord with our expectations we obtain that the high-frequency oscillations are absent in the dynamics of $\rho$, while the low-frequency oscillations are still present. On the other hand, the low-frequency oscillations vanish in the $\Sigma$ transient in contrast to the high-frequency components. This becomes even more clear from the fast Fourier transform (FFT) spectra of $\rho$ and $\Sigma$ shown in Fig. 2c. From the fit of the $\rho$ transient we evaluate $\delta_1 = 0.25$ meV and $\delta_2 = 0.55$ meV, which are in agreement with the exciton fine structure splitting evaluated by two-dimensional Fourier spectroscopy on CsPbI$_3$ NCs of similar size: cube-shaped with side length of about 9 nm[46]. A detailed discussion of the fine exciton structure combining pump-probe and PE studies will be published elsewhere. In what follows, we focus on the dynamics of $\Psi_0(\tau_{12})$, which is independent of the spin level structure of the exciton.

The FFT spectrum of $\Sigma$, shown by the yellow line in Fig. 2c, exhibits several spectrally narrow features. These features are consistent with the peaks in the Raman spectrum in Fig. 2d. The latter originates from light scattering on optically active phonons. The features at energies below 0.5 meV are attributed to confined acoustic phonons[70]. Below we focus on the two most prominent features marked by the vertical dashed lines with energies of $\hbar\Omega_1 = 3.2$ meV and $\hbar\Omega_2 = 5.1$ meV, corresponding to the energies of optically active optical phonons in the vicinity of the $\Gamma$ point. Indeed optical phonon modes with energies close to 3 and 5 meV in CsPbI$_3$ were observed in ref. 23 and calculated in ref. 8. Optical phonons with similar energies were also detected by Raman spectroscopy in other lead halide perovskites such as FAPbI$_3$[71,72], CsPbBr$_3$[21], and CsPbCl$_3$[19,73]. Thus, we conclude that the high frequency THz oscillations in the two-pulse coherent optical response are due to the coherent evolution of the coupled, hybridized exciton-phonon system.

## Exciton-polaron quantum beats

In order to describe the quantum dynamics of the coupled exciton-phonon system we consider the structure of the optically excited energy levels in a single NC, which requires the involvement of exciton-polaron states. To that end, we introduce the four-level scheme shown in Fig. 3b which comprises the state with zero excitons and zero phonons in the NC $|0\rangle$, the state with zero excitons and one optical phonon $|0'\rangle$, the ground state exciton-polaron (i.e., the state with one exciton and no phonons, but with account of lattice relaxation) $|X\rangle$, and the state with one exciton and one phonon $|X'\rangle$ (excited exciton-polaron state, see details in the Supplementary Note 5). Due to the relatively small Huang-Rhys factor one may neglect the modes with

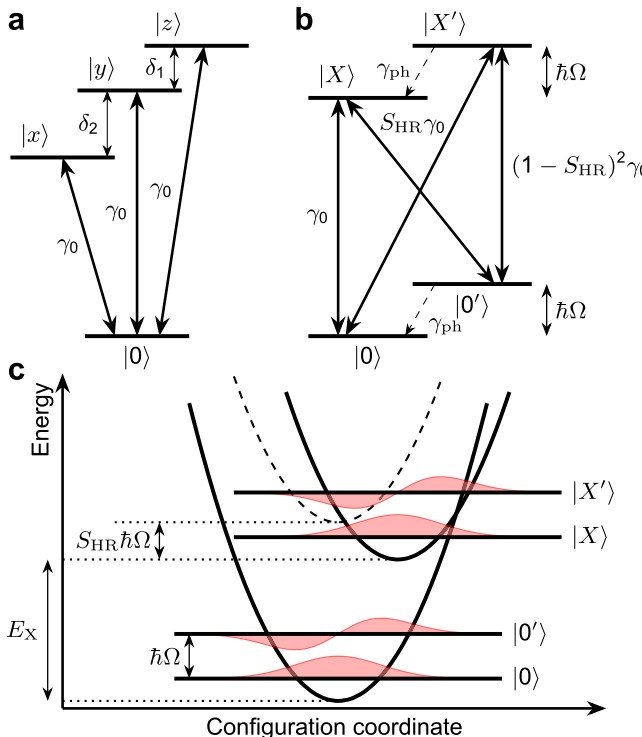

**Fig. 3 | Energy level schemes. a** Energy levels of the bright exciton fine structure. **b** Energy level structure of the excitons interacting with optical phonons. We assume this level structure for each phonon mode, independent of the exciton polarization. The probability of the diagonal transitions is proportional to the Huang-Rhys factor $S_{\mathrm{HR}}$. $\gamma_{\mathrm{ph}}$ is the phonon decay rate and $\gamma_0$ is the exciton decay rate, dominated by the radiative decay. **c** Configurational coordinate diagram for a given phonon mode with adiabatic potentials in the ground nanocrystal (NC) state and a NC state with an exciton shown as solid parabolas. The dashed parabola indicates the adiabatic potential for a NC state with an exciton in the case when the exciton-phonon coupling is neglected. The solid parabolas are separated by the energy difference $E_X$ corresponding to the energy of zero-phonon optical transition. $\hbar\Omega$ denotes the energy of the optical phonon.

many phonons. We neglect the exciton fine structure splitting to concentrate on the derivation of $\Psi_0(\tau_{12})$ in Eq. (1). Initially, we consider the exciton coupling to a single phonon mode in a NC. At the next stage, to compare with the experimental data, we perform summation over different phonon modes $\Omega_i$ where we neglect the interaction between them

$$\Psi_0(\tau_{12}) = \sum_i \Psi_0(\tau_{12}; \Omega_i). \tag{4}$$

The optical transitions between all energy levels are allowed using the same polarization. In the exciton-polaron model[74,75], the transition probability amplitude is proportional to the product of the dipole matrix element and the overlap of the wavefunctions associated with vibronic modes shifted due to polaron formation (see Fig. 3c and Supplementary Note 5), which depends on the strength of the exciton-phonon interaction given by the Huang-Rhys factor $S_{\mathrm{HR}}$ (Condon principle). The pulse duration is assumed to be short as compared with all other characteristic times in the system, i.e., we can assume the $\delta$-pulse limit. This is justified because the laser pulse duration is $\tau_d \approx 120$ fs (see "Methods") and the fit to experimental data gives relaxation times in the order of 10 ps. The spectral width of the laser pulse $\approx 0.44\hbar/\tau_d$ is about 15 meV, which is larger than the phonon energies $\hbar\Omega_i$. Therefore, we assume that all four transitions between states $|0\rangle$, $|0'\rangle$ and $|X\rangle$, $|X'\rangle$ are covered by the spectral width of the laser.

Although multiphonon states (e.g., $2\hbar\Omega_1$) lie within the laser spectral bandwidth, they can be neglected owing to the relatively small Huang-Rhys factor. We stress that the observation of pronounced quantum beats and the applicability of the model strongly relies on the long exciton coherence times $T_2$, well resolved phonon frequencies, and intermediate Huang-Rhys factors. The choice of material is particularly important because in bromides, the optical phonon modes are relatively broad[22,76], which may lead to rapid dephasing of excited exciton-polaron states.

The anharmonicity of optical phonon modes, critically important at room temperature[30,52,77], is known to have a minor effect at cryogenic temperatures[72]. Even though the signal contains peaks at double phonon frequencies, this is an internal feature of nonlinear FWM, being present in the optical response of the sample due to interference between the $0' \to X$ and $0 \to X'$ transitions (see Supplementary Note 5). Also, we neglect the possible difference of phonon energy in the crystal ground and optically excited states as we do not find corresponding frequency changes in our experimental data and in previous reports in other perovskites[71,72]. Similar assumptions were made in the model developed for describing 2D spectroscopy in ref. 78.

In the exciton-polaron model, we trace the evolution of the density matrix components which contribute to the PE. The result arises from the coherent off-diagonal terms and all levels contribute to the result in first order in the Huang-Rhys factor $S_{\mathrm{HR}}$, which quantifies the exciton-phonon interaction. Solution of the Lindblad equation in Supplementary Note 5 gives the following expression for the amplitude of the phonon echo signal from a NC with single phonon mode $\Omega_i$

$$\Psi_0(\tau_{12}; \Omega_i) = e^{-(1+S_{\mathrm{HR}})\gamma_0\tau_{12}} \Big[ 1 + S_{\mathrm{HR}} e^{-(\gamma_{\mathrm{ph}}-S_{\mathrm{HR}}\gamma_0)\tau_{12}}$$
$$+ S_{\mathrm{HR}} \cos(\Omega_i\tau_{12}) e^{-\frac{\gamma_{\mathrm{ph}}-2S_{\mathrm{HR}}\gamma_0}{2}\tau_{12}} \Big[ 2 + e^{-S_{\mathrm{HR}}\gamma_0\tau_{12}} + e^{-\gamma_{\mathrm{ph}}\tau_{12}} \Big] \tag{5}$$
$$+ S_{\mathrm{HR}} \cos(2\Omega_i\tau_{12}) e^{-(\gamma_{\mathrm{ph}}-S_{\mathrm{HR}}\gamma_0)\tau_{12}} \Big].$$

Here $T_{2,0} = 1/\gamma_0$ is the coherence time associated with the zero-phonon optical transition, and $\tau_{\mathrm{ph}} = 1/\gamma_{\mathrm{ph}}$ is the lifetime of the optical phonon. In the simplified model we do not include other sources of decoherence. The analytical expression fully supports the experimental observations for $\Sigma = \Psi_0(\tau_{12})$ shown in Fig. 2b. The first term in Eq. (5) corresponds to the long-lived zero-phonon coherence which decays exponentially with the time $T_2 = T_{2,0}/(1 + S_{\mathrm{HR}}) \sim 300$ ps, as evaluated above. The last two terms correspond to high frequency oscillations at the single and double frequency of the phonon mode $\Omega_i$, respectively. They appear due to quantum interference of excitations of different exciton-polaron states, i.e., due to quantum beats of exciton-polarons. These oscillations are superimposed on the long-lived signal and decay with a shorter time $\tau_{\mathrm{ph}}$. The relative amplitude of the oscillatory signal and the long-lived plateau allows one to measure the Huang-Rhys factor $S_{\mathrm{HR}}$.

Using Eqs. (4) and (5) for two independent phonon modes with $\hbar\Omega_1 = 3.2$ meV and $\hbar\Omega_2 = 5.1$ meV we obtain excellent agreement with the experimental data. The phonon frequencies are taken from the peak positions in the Fourier spectrum of Fig. 2c. The Huang-Rhys factors and phonon lifetimes are evaluated from the best fit to the experimentally measured transients, see Fig. 2b. We emphasize that in contrast to previous reports[39–43], our system demonstrates long-lived coherent dynamics, which is attributed to an exceptionally long zero-phonon exciton coherence ($\sim$300 ps) and a relatively long phonon lifetime ($\sim$10 ps). Here, we stress that the low-temperature regime with $T = 2$ K is essential, as the coherence of the zero-phonon excitons rapidly vanishes with increasing temperature. Furthermore, the optical phonons each are represented by well-defined frequencies due to their flat dispersion around the Γ-point, as confirmed by the Raman spectrum in Fig. 2d, making CsPbI₃ NCs embedded in a glass matrix particularly interesting for investigating coupled exciton-phonon dynamics. The peak widths in the Raman spectrum can be recalculated

into phonon lifetimes using the following relation $\tau_{ph} = 2\hbar/\Delta E_R$[79], where $\Delta E_R$ is the full width at half maximum in energy units. This yields phonon lifetimes of ~7 ps and 11 ps for the 3.2 and 5.1 meV modes, respectively. These values are in excellent agreement with the decay times of the PE oscillatory signals. It should be noted that the Raman peaks in Fig. 2d appear narrower than those in the FFT spectra in Fig. 2c, which originates from the difference between the power spectral and the amplitude spectral representations, respectively.

The proposed exciton-polaron model (i) provides an intuitive picture of the microscopic processes involved, (ii) has a simple analytical solution, and (iii) accounts explicitly for the coherence decay through simple equations. Equation (5) gives a result that closely matches that in refs. 39,80 and is in effect equivalent within the accuracy limits of the models. Note that a direct comparison between the results obtained by the two approaches is not straightforward: Eq. (5) is obtained from the exact solution of the Lindblad equations for a four-level system, while ref. 80 considers the quasi-classical evolution of a complex system averaged over the phonon subsystem. For a detailed discussion and comparison of the two approaches see e.g., ref. 81.

## NC size dependence

Above, we discussed the PE dependence measured for the fixed laser pulse photon energy of $h\nu = 1.746$ eV and explained how the phonon parameters can be evaluated from the PE signal. However, the sample under study contains NCs with different sizes which can be selectively excited by tuning the laser photon energy. We measured PE transients at photon energies in the range of $1.72 - 1.76$ eV. The photon energy can be recalculated into a NC diameter using the empirical fit $D = \sqrt{16.93/(E_X - 1.652)} - 4.31$ (where $E_X$ is the position of exciton peak in eV) to the results from ref. 51, where the same sample was studied (sample #3, see also Supplementary Note 2). Figures 4a, b show the dependence of the evaluated Huang-Rhys factors $S_{HR}$ and phonon lifetimes $\tau_{ph}$ as function of the NC diameter, for phonon modes with energies $\hbar\Omega_1 = 3.2$ meV and $\hbar\Omega_2 = 5.1$ meV. We note that the frequencies of these modes do not depend on NC size within the accuracy of the experiment (see Supplementary Note 6). The Huang-Rhys factors for both phonon modes exhibit a clear size dependence, increasing with decreasing NC size. Correspondingly, the phonon lifetimes decrease with decreasing $a$. An increase in the electron-phonon coupling strength with decreasing NC size was previously reported for CdSe nanocrystals[82], PbS quantum dots[83], and perovskite nanocrystals, both for optical and acoustic phonons[18,21]. However, to the best of our knowledge, direct measurements of the associated phonon relaxation times have not been reported. It is also important to note that, in contrast to previous studies, our PE experiments selectively probe the coherent dynamics of each individual phonon mode.

In ref. 21, the dominant mechanism of electron-phonon coupling is assigned to the optical deformation potential[84]. In this case, the size dependence of the interaction can be estimated from the phonon normalization condition[7] which leads to $S_{HR} \sim a^{-3}$, where $a$ is the NC radius. For completeness, let us discuss the size dependence of the interaction between charge carriers and optical phonons for the Fröhlich mechanism following Takagahara[85]. In the original work, the interaction of charge carriers with phonons is rewritten as potential for an electron in the field induced by the phonon mode polarization $\mathbf{P}(\mathbf{r})$. This potential is found from the Poisson equation $\nabla^2\varphi(\mathbf{r}) = 4\pi\nabla \cdot \mathbf{P}(\mathbf{r})$ which leads to

$$\varphi(\mathbf{r}_e) = \int d\mathbf{r}\, \frac{\nabla \cdot \mathbf{P}(\mathbf{r})}{|\mathbf{r} - \mathbf{r}_e|} = -\int d\mathbf{r}\, \mathbf{P}(\mathbf{r}) \cdot \nabla \frac{1}{|\mathbf{r} - \mathbf{r}_e|}. \quad (6)$$

From Eq. (6) the strength of electron-phonon interaction $\Delta_e \sim a^{-1}$ which leads to $S_{HR}^e = \Delta_e^2/2 \sim a^{-2}$. This estimate is valid also for excitons in the strong confinement regime.

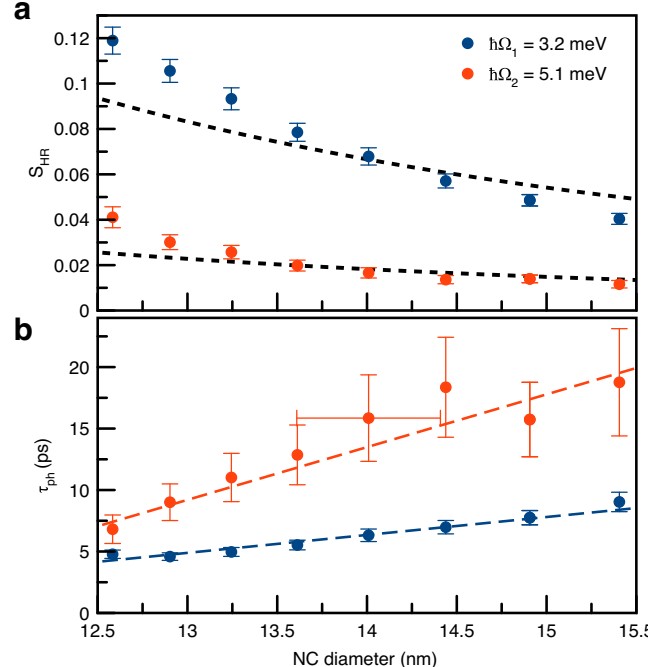

**Fig. 4 | Nanocrystal (NC) size dependence.** Dependence of **a** the Huang-Rhys factor, $S_{HR}$, and **b** the phonon lifetime, $\tau_{ph}$, for the two phonon modes with energies $\hbar\Omega_1 = 3.2$ meV (blue dots) and $\hbar\Omega_2 = 5.1$ meV (red dots) on NC diameter. Dashed lines in a are fits proportional to $a^{-3}$. Dashed lines in b are guides to the eye. The horizontal error bar represents the uncertainty associated with the spectral width of the PE signal, corresponding to $\pm 5$ meV. Vertical bars represent standard deviation.

For weakly confined excitons, the change of exciton energy $\Delta$ is proportional to the difference of the electrostatic potential for electron and hole $\varphi(\mathbf{r}_e) - \varphi(\mathbf{r}_h)$. It is found to be the sum of Eq. (6) for electron and hole:

$$\Delta \sim \varphi(\mathbf{r}_e) - \varphi(\mathbf{r}_h) = \int d\mathbf{r}\, \mathbf{P}(\mathbf{r}) \cdot \nabla \left( \frac{1}{|\mathbf{r} - \mathbf{r}_h|} - \frac{1}{|\mathbf{r} - \mathbf{r}_e|} \right). \quad (7)$$

In the weak confinement regime when the exciton Bohr radius $a_B$ is small compared with the NC radius $a_B \ll a$,

$$\frac{1}{|\mathbf{r} - \mathbf{r}_h|} \approx \frac{1}{|\mathbf{r} - \mathbf{r}_e|} + (\mathbf{r}_e - \mathbf{r}_h) \cdot \nabla \frac{1}{|\mathbf{r} - \mathbf{r}_e|}. \quad (8)$$

Substituting Eq. (8) into Eq. (7) we obtain

$$\varphi(\mathbf{r}_e) - \varphi(\mathbf{r}_h) \approx \int d\mathbf{r}\, \mathbf{P}(\mathbf{r}) \cdot \nabla \left[ (\mathbf{r}_e - \mathbf{r}_h) \cdot \nabla \frac{1}{|\mathbf{r} - \mathbf{r}_e|} \right]. \quad (9)$$

As a result, in the weak confinement regime, the size dependence of this expression is estimated as

$$\Delta \sim \varphi(\mathbf{r}_e) - \varphi(\mathbf{r}_h) \propto a^3 \cdot a^{-3/2} \cdot \frac{a_B}{a^3} \propto a^{-3/2}. \quad (10)$$

Here, the first size factor comes from the integral, the second one from the normalization of the phonon mode, and the last one from the gradient of the expression in the square brackets. As a result the Huang-Rhys factor scales as $S_{HR} = \Delta^2/2 \sim a^{-3}$.

Thus, in the weak confinement regime both mechanisms, the deformation potential and the Fröhlich interaction, lead to similar dependences of the Huang-Rhys factor on the NC size $S_{HR} \sim a^{-3}$. In the

strong confinement regime, the Fröhlich interaction is expected to have a weaker size dependence, while the deformation potential mechanism should show the same scaling. The dashed lines in Fig. 4 represent fits of the experimental data by the empirical relation $S_{HR} \propto a^{-3}$. The experimental data in Fig. 4a even shows a steeper dependence of the Huang-Rhys factor on NC size, with pronounced deviation from the $a^{-3}$ dependence, particularly for small NCs. This observation suggests that further analysis is required for smaller NCs to identify the mechanisms of electron-phonon interaction in perovskite NCs.

## Discussion

To summarize, we have revealed a fully coherent exciton-polaron dynamics following resonant optical excitation with femtosecond pulses in an ensemble of CsPbI$_3$ nanocrystals embedded in a glass matrix. Quantum beats between exciton-polaron states manifest themselves in two-pulse photon echo experiments as oscillations with high frequencies corresponding to two optical phonon modes at 3.2 and 5.1 meV. A polarization resolved analysis of the photon echo signal has allowed us to unambiguously isolate the quantum beat signatures of exciton-polarons from slower beats arising from exciton fine-structure splittings. This unique regime of exciton-polaron quantum beats is established at the low temperature of 2 K, where an exceptionally long zero-phonon optical coherence time is achieved ($T_2 \approx 300$ ps). The presence of spectrally narrow optical phonon modes with lifetimes of about 10 ps, combined with a relatively strong exciton-phonon coupling (Huang-Rhys factors up to 0.1), leads to a pronounced modulation of the photon echo signal at the phonon frequencies. Furthermore, the strong size dependence of the Huang-Rhys factor makes it possible to tune the strength of the optical transitions between different exciton-polaron states, offering a route to control the quantum dynamics for targeted applications.

We emphasize that the observed optical response markedly differs from previous studies in, e.g., III-V self-assembled quantum dots, where the dynamics is mainly governed by exciton-acoustic phonon coupling[27,86,87]. There, the continuous phonon spectrum leads to a partial loss of coherence within the first few picoseconds after pulsed excitation. In our case no such behavior is observed in the temporal evolution of $\Psi_0(\tau_{12})$ [see Fig. 2b]. Instead, the coherent dynamics is dominated by exciton-polaron states, i.e., interactions with discrete optical phonon modes, which can be consistently described within a four-level model for each phonon frequency. The desired quantum evolution can be achieved by tuning both the exciton-phonon coupling strength and, most importantly, the phonon lifetime, which limits the evolution of the pure quantum state following resonant excitation with a short optical pulse. Notably, the phonon lifetime exhibits a pronounced dependence on nanocrystal size. For the optical phonon mode with an energy of 5.1 meV, we observe an almost twofold increase of $\tau_{ph}$ as the NC diameter increases from 12 to 15 nm. In layered perovskites, phonon lifetimes on the order of ~10 ps were reported[88], with values extended up to 75 ps[89], approaching the record-high ~200 ps observed in bulk ZnO[79]. Therefore, the strong size dependence of phonon lifetimes, along with reports of extended phonon coherence in layered perovskites, gives a promising outlook for long-lived and tunable coherent control of the exciton-polaron dynamics in perovskite nanocrystals.

## Methods
### Samples
The studied CsPbI$_3$ nanocrystals embedded in fluorophosphate Ba(PO$_3$)$_2$-AlF$_3$ glass were synthesized by rapid cooling of a glass melt enriched with the components needed for the perovskite crystallization. Details of the method are given in refs. 50,90,91. The data on NC sizes evaluated from scanning transmission electron microscopy can be found in the supporting information section S5 in ref. 51. Samples of fluorophosphate (FP) glass with the composition 35P$_2$O$_5$–35BaO–5AlF$_3$–10Ga$_2$O$_3$–10PbF$_2$–5Cs$_2$O (mol. %) doped with BaI$_2$ were synthesized using the melt-quench technique. The glass synthesis was performed in a closed glassy carbon crucible at the temperature of $T = 1050$°C. Technological code of the studied sample is EK8. More information on its optical and spin-dependent properties can be found in refs. 51,92,93.

The sample is placed in the variable temperature insert of a bath cryostat and kept in superfluid helium at the temperature of 2 K. The low energy exciton optical transitions in nanocrystals are located in the spectral range of 1.7–1.8 eV as follows from the photoluminescence spectrum, see Fig. 1a.

### Photon echo
Two-pulse photon echoes were measured using a transient four-wave mixing technique with heterodyne detection[56–59]. We used a Ti:Sapphire laser as source of optical pulses with duration of 120 fs at the repetition rate of 75.75 MHz. Some of the measurements were performed with spectrally narrow 0.6 meV wide laser pulses with a duration of about 3 ps. All optical experiments were carried out at photon energies below 1.76 eV. The sample is excited resonantly by a sequence of two pulses separated in time by the delay $\tau_{12}$, giving rise to the formation of a four-wave mixing signal with electric field amplitude $\mathcal{E}_{FWM}(t)$. The pulses are focused into a spot of about 200 $\mu$m diameter using a 0.5-m parabolic aluminum mirror. The incidence angles of the pulses are close to normal and equal to ~1/50 rad and 2/50 rad, corresponding to the wavevectors $\mathbf{k}_1$ and $\mathbf{k}_2$. An additional reference pulse is used for detection and its delay with respect to the first pulse is given by the time $\tau_{ref}$. The delay times $\tau_{12}$ and $\tau_{ref}$ are controlled using mechanical delay lines. The four-wave mixing signal is measured in transmission geometry in the direction $\approx 3/50$ rad, which corresponds to the phase-matching direction $2\mathbf{k}_2 - \mathbf{k}_1$. Optical heterodyne detection is used to resolve the enhanced signal in time. By mixing with a relatively strong reference pulse and scanning $\tau_{ref}$, the temporal profile of the photon echo pulse is measured.

Polarization sensitive measurements are performed by selecting proper polarization configurations (H-horizontal or V-vertical) for each of the beams using Glan-Thompson prism in conjunction with half-wave plates. In the detection path we also select the linear polarization by setting the polarization of the reference beam. We adopt a three-letter notation for the PE polarization configuration. The first two symbols correspond to the polarizations of the pulses exciting the sample, and the third symbol corresponds to the detection polarization. For example, in the HHH configuration, the polarizations of both excitation pulses and the detection are horizontal (H). In the HVH configuration, the second excitation pulse is vertically (V) polarized, while the first pulse and detection are horizontally polarized. HHH is accordingly termed co-linearly polarized (∥) and HVH cross-linearly polarized (×) in Fig. 2. We note that rotation of the sample does not lead to changes in the signal, i.e., the orientation of the sample is not important.

### Raman spectroscopy
The Raman scattering signal was excited by a single-frequency Ti:Sapphire laser. The laser beam with the power of about 1 mW was focused on the sample to a spot with a diameter of about 300 $\mu$m. The scattered light was analyzed by a Jobin-Yvon U1000 double monochromator with 1-meter focal length, allowing the high resolution of 0.2 cm$^{-1}$ (0.024 meV). The Raman signal was detected by a cooled GaAs photomultiplier and conventional photon-counting electronics. The Raman spectra were measured in co-polarized linear polarizations of excitation and detection.

## Data availability

The authors declare that the main data supporting the findings of this study are available within the article and its Supplementary Information files. Raw data are available under accession code (https://doi.org/10.6084/m9.figshare.31985061). Extra data are available from the corresponding authors upon request.

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

## Acknowledgements
The authors are thankful to J. Even, I. A. Yugova and N. E. Kopteva for fruitful discussions.

## Author contributions
A.V.T., M.-A.H. and S.G. performed the photon echo experiment. D.K. performed Raman spectroscopy. M.O.N., S.V.G., J.M.K. and D.E.R. developed the theoretical model. E.V.K. and M.S.K. fabricated the samples. A.V.T., M.O.N. and I.A.A. conceived the experiment and carried out the analysis. A.V.T., M.O.N., I.A.A., D.R.Y. and M.B. cowrote the paper. All authors discussed the results and commented on the manuscript.

## Funding
We acknowledge the financial support by the Deutsche Forschungsgemeinschaft: A.V.T., M.A.H., S.G., M.O.N. and I.A.A. (project AK40/13-1, no. 506623857) and D.R.Y. (project YA65/28-1, no. 527080192). The work of S.V.G. was supported by the NSF through DMR-2100248. A.V.T., E.V.K. and M.S.K. acknowledge the Saint-Petersburg State University (Grant No. 125022803069-4). Open Access funding enabled and organized by Projekt DEAL.

## Competing interests
The authors declare no competing interests.
