## [Transparent Peer Review file · Nature Communications]

Quantum beats of exciton-polarons in CsPbI₃ perovskite nanocrystals

Corresponding Author: Dr Ilya Akimov

Version 0:

Reviewer comments:

Reviewer #1

(Remarks to the Author)

In this manuscript, Trifonov and coworkers reported the observation of quantum beats of exciton-polarons in CsPbI₃ perovskite nanocrystals embedded in glass using transient two-pulse photon echo spectroscopy. The observation is distinct in that previous photon echo studies on perovskite nanocrystals reported exciton coherence time T₂ but they didn't report simultaneous phonon coherence, whereas other studies extensively reported coherent phonon dynamics but not in the exciton coherence regime (i.e., incoherent exciton dynamics modulated at the frequencies of phonons). By contrast, the present study reported coherent exciton-polaron quantum beats well within the exciton T₂. This is partially enabled by the very long exciton coherence at 2 K, with T₂ as long as 330 ps. The observation and analysis here are definitely important and interesting. I would recommend the publication of this manuscript in Nat. Commun., if the authors can address the minor issues listed below.

1. As mentioned above, the uniqueness of this study is the observation of phonon coherence well within the exciton T₂. However, it is unclear what makes the present system unique for this observation. In other words, why are the phonon coherence missing in previous studies, such as refs 28, 37. The samples are indeed different, nanocrystals in glass versus colloidal nanocrystals, but this should not change the strength of electron-phonon coupling which is dictated by the intrinsic properties of the perovskite lattice.
2. Related to the comment above, there indeed exists a previous study in which the exciton fine-structure coherence (not equivalent to exciton coherence, though) coexist with phonon coherence. In Nat. Mater. 2022, 21, 1282–1289, Han reported simultaneous observation of exciton fine-structure coherence and phonon coherence. The phonon modulation is very shallow, but it can be observed at high pump fluences (Supplementary Fig 4 therein) and the phonon beating frequency (about is also much faster than the fine-structure quantum beating. The authors should discuss the distinction of the present study from that work.
3. The sensitivity of the exciton T₂ to the nanocrystal size seems to be too sensitive. The authors basically select different exciton energies from the ensemble by tuning the laser frequencies. But the 100 fs pulse itself is very broad, so the energy selection is not that accurate. In this case, it is hard to imagine that in such a small energy range (1.724 to 1.75 eV), T₂ varies from 330 to 230 ps. How accurate are the fitting parameters? Error bars should be given.
4. The authors may consider the anharmonicity effect in the observed the phonon dephasing dynamics. The method is well documented in the literature (e.g., J. Phys. Chem. Lett. 2022, 13, 5413–5423). It would be interesting to compare the anharmonicity of the two optical phonon modes at 5.1 and 3.2 meV, as well as the effect of nanocrystal size on this anharmonicity.

Reviewer #2

(Remarks to the Author)

This manuscript reports on exciton-lattice interactions in perovskite QD, specifically CsPbI₃. The topic itself is of paramount importance and is a major question in quantum materials. But there have been many efforts to address this topic using femtosecond and single dot PL spectroscopies. So it must be clear in the paper what is new in 2026. That point is far from obvious in this paper. It is an excellent paper. The data is lovely. They writing is great. But what is new? Moreover some of the main concepts of lattice dynamics are not correct. Details are below.

1. The abstract and title move between phonons and polarons. Both are relevant and care must be taken to relate the two. Phonons exist at 2K as normal modes that are coherent. Polarons exist at 300 K as local modes from the basis of normal modes, and are diffusive. They are related but not the same. This point must be addressed up front.

2. The abstract discusses the importance of lattice dynamics but in the end only measures exciton-phonon coupling strength. More should be shown.
3. The authors review the literature on phonons in perovskite QD. Some are cited here(1-10). So what is new?
4. The authors discuss the long lived electronic coherence of the exciton. But is this really long-lived coherence? IN 2DE and CMDS, long-lived coherence means the electronic coherence between excitonic states is long lived relative to some internal clock such as the T2 of the basis excitons. Such as case has been found for inter exciton coherence at 300 K(5).
5. The synthetic methods of the QD are not the state of the art as shown by other authors such as Kovalenko or DH Son. As a result great care must be taken in the characterization of them. There must be much information about them in the SI.
6. What is the 2K T2 compared to single dot experiments by Bawendi(11)?
7. The most interesting FFT signal is the zero frequency rolloff. It is not seen in Raman because CW methods cannot measure low frequency accurately but time domain can. So what is this roll off? Is that the residual thermal disorder of pre polarons at 2K?
8. Why is the coupling, S, closer to 0.1 instead of 1 as seen in other measurements.

1. S. Biswas et al., Exciton polaron formation and hot-carrier relaxation in rigid Dion–Jacobson-type two-dimensional perovskites. *Nature Materials*, 1-7 (2024).
2. X. Yue et al., Real-time observation of the buildup of polaron in α -FAPbI₃. *Nat. Commun.* 14, 917 (2023).
3. C. D. Sonnichsen, D. P. Strandell, P. J. Brosseau, P. Kambhampati, Polaronic quantum confinement in bulk CsPbBr₃ perovskite crystals revealed by state-resolved pump/probe spectroscopy. *Physical Review Research* 3, 023147 (2021).
4. F. Thouin et al., Phonon Coherences Reveal the Polaronic Character of Excitons in Two-Dimensional Lead Halide Perovskites. *Nat. Mater.* 18, 349 (2019).
5. A. Ghosh et al., Correlated Lattice Fluctuations in CsPbBr₃ Quantum Dots Give Rise to Long-Lived Electronic Coherence. *ACS Nano* 19, 19927-19937 (2025).
6. O. Yaffe et al., Local Polar Fluctuations in Lead Halide Perovskite Crystals. *Phys. Rev. Lett.* 118, 136001 (2017).
7. C. Zhu et al., Quantifying the size-dependent exciton-phonon coupling strength in single lead-halide perovskite quantum dots. *Advanced Optical Materials* 12, 2301534 (2024).
8. K. Cho et al., Exciton–phonon and trion–phonon couplings revealed by photoluminescence spectroscopy of single CsPbBr₃ perovskite nanocrystals. *Nano Lett.* 22, 7674-7681 (2022).
9. O. Pfingsten et al., Phonon interaction and phase transition in single formamidinium lead bromide quantum dots. *Nano Lett.* 18, 4440-4446 (2018).
10. M. Fu et al., Unraveling exciton–phonon coupling in individual FAPbI₃ nanocrystals emitting near-infrared single photons. *Nat. Commun.* 9, 3318 (2018).
11. H. Utzat et al., Coherent Single-Photon Emission from Colloidal Lead Halide Perovskite Quantum Dots. *Science* 363, 1068 (2019).

Reviewer #3

(Remarks to the Author)

This manuscript reports the observation of exceptionally long coherence times of exciton-polarons in CsPbI₃ nanocrystals, which is interesting and novel in quantum physics. The underlying mechanism is analyzed through a profound size dependent theoretical calculation. This study can be accepted if the following questions are addressed:

- 1) In Fig.1 a. one may observe multi-peaks configuration, what's the reason of occurrence of these peaks? If they origin from nanocrystals with different sizes, considering the size will affect the decay times of the phonons, a detailed analysis or supporting data should be made to clarify the size effect.
2. For nanocrystals with different sizes, the same phonon energies $\hbar\Omega_1 = 3.2$ meV and $\hbar\Omega_2 = 5.1$ meV are used however, this is not necessarily the case, the phonon energies of different nanocrystals should be investigated.
3. In the main text the author wrote "making CsPbI₃ NCs embedded in a glass matrix particularly interesting for investigating coupled exciton-phonon dynamics." But they did not clarify how the glass matrix impacts the exciton-phonon coupling ? Is this glass matrix crucial for the observation of the long coherent times of exciton polaron?
4. The exciton coherent time is dependent on the photon energy, why?
5. In ref[15][*Nat. Mater.* 18, 717–724 (2019)], the bright-dark exciton splitting would be determined by the crystal local symmetry and quantum confinement, I suggest the consideration of this effect for the discussion of the exciton-polarons.

Version 1:

Reviewer comments:

Reviewer #1

(Remarks to the Author)

The authors have addressed my and other referee's previous comments. The paper can be accepted for publication.

Reviewer #2

(Remarks to the Author)

I have now read the revised manuscript and the authors' response carefully. In my view, the paper is now in publishable shape and can reasonably be accepted after minor revision.

Overall, I find this to be an interesting and worthwhile contribution. The central experimental result is real and significant: the authors report unusually long zero-phonon optical coherence in glass-embedded CsPbI

3
3

nanocrystals at cryogenic temperature, together with resolved quantum beats arising from low-energy exciton–phonon structure. In my opinion, this is the true strength of the work. The paper is valuable not simply because oscillations are observed, but because the authors appear to access a coherent-optical regime in which the optical coherence survives long enough that the coupled exciton–phonon manifold becomes directly visible in the photon-echo response. That is a nontrivial result and will be of interest to researchers working in coherent spectroscopy, perovskite nanocrystals, and exciton–phonon coupling.

I also think the revised manuscript is improved in important ways. The authors have answered most of the substantive technical questions reasonably well, and the paper is now more precise about what is, and is not, being claimed. In particular, it is helpful that the authors distinguish the unusually long zero-phonon optical coherence observed in their experiment from the broader notion of long-lived inter-state coherence often discussed in the multidimensional-spectroscopy literature. That clarification makes the paper more accurate and considerably stronger.

My remaining concern is not about the quality of the experiment, but about terminology and conceptual scope. I still think one should be careful with the term polaron at 2 K. In the stronger sense usually associated with dynamical lattice reorganization or real-time polaron formation, I would be cautious about over-interpreting the present data. To my mind, the safer language is that the authors observe phonon-dressed excitonic states in a regime of unusually long optical coherence. That said, I do not regard this as a reason to delay publication. Rather, I would suggest a minor further tightening of language so that the manuscript consistently makes clear that its use of “exciton-polaron” refers to a dressed confined excitonic state, not necessarily to the stronger claim of finite-temperature dynamical polaron birth.

In addition, I think the manuscript would benefit from somewhat broader engagement with the recent coherent multidimensional spectroscopy literature on perovskite quantum dots, especially work bearing on homogeneous lineshape dynamics, correlated lattice fluctuations, coherence, and polaronic interpretation in PQDs. The present paper will sit more naturally and more convincingly within the field if it acknowledges more fully that evolving body of work. In particular, I believe the authors should consider citing the following relevant studies:

Seiler, H. et al. Two-dimensional electronic spectroscopy reveals liquid-like lineshape dynamics in CsPbI

3
3

perovskite nanocrystals. *Nature Communications* 2019, 10, 4962.

Ghosh, A. et al. Correlated lattice fluctuations in CsPbBr

3
3

quantum dots give rise to long-lived electronic coherence. *ACS Nano* 2025, 19, 19927–19937.

Ghosh, A. et al. Coherent multidimensional spectroscopy reveals homogeneous lineshape dynamics in CsPbBr

3
3

quantum dots. *ACS Nano* 2025, 19, 26843–26851.

Nagpal, P.; Kambhampati, P. Real-Time Formation of a Landau Polaron. *arXiv* 2026, 2602.24113.

I am not suggesting that the paper be reframed around these works, only that it would be strengthened by recognizing more explicitly the recent CMDS/2DES literature on PQDs and by locating the present result within that broader context.

So my recommendation is straightforward: accept after minor revision. The remaining revisions, in my opinion, should be limited to:

ensuring terminological precision around the use of “exciton-polaron” at 2 K, and

strengthening the citation context with respect to recent CMDS/2DES work on perovskite quantum dots.

With those minor adjustments, I believe the paper will make a valuable contribution to the literature.

Reviewer #3

(Remarks to the Author)

The revised manuscript has replied most of my concerns, however, the distribution of the size of the nanocrystals is still lacking, I suggest the authors provide the microscopic images and analysis of the sizes as these are crucial for the explanation of their data.

Version 2:

Reviewer comments:

Reviewer #3

(Remarks to the Author)

The author has response to my comments, and I think the manuscript is ready for publication.

Point-by-point response to the reviewers' comments

Response to Reviewer #1 (Remarks to the Author):

Rev1: In this manuscript, Trifonov and coworkers reported the observation of quantum beats of exciton-polarons in CsPbI₃ perovskite nanocrystals embedded in glass using transient two-pulse photon echo spectroscopy. The observation is distinct in that previous photon echo studies on perovskite nanocrystals reported exciton coherence time T₂ but they didn't report simultaneous phonon coherence, whereas other studies extensively reported coherent phonon dynamics but not in the exciton coherence regime (i.e., incoherent exciton dynamics modulated at the frequencies of phonons). By contrast, the present study reported coherent exciton-polaron quantum beats well within the exciton T₂. This is partially enabled by the very long exciton coherence at 2 K, with T₂ as long as 330 ps. The observation and analysis here are definitely important and interesting. I would recommend the publication of this manuscript in Nat. Commun., if the authors can address the minor issues listed below.

R1: We thank the reviewer for the stimulating questions and constructive comments, which have helped us to improve the manuscript. We address all points below.

Q1.1. *As mentioned above, the uniqueness of this study is the observation of phonon coherence well within the exciton T₂. However, it is unclear what makes the present system unique for this observation. In other words, why are the phonon coherence missing in previous studies, such as refs 28, 37. The samples are indeed different, nanocrystals in glass versus colloidal nanocrystals, but this should not change the strength of electron-phonon coupling which is dictated by the intrinsic properties of the perovskite lattice.*

R1.1. We thank the reviewer for this comment and pointing out these references.

The main difference is that all previous studies on coherent optical spectroscopy (Refs. [35,44] in the revised version) were performed on ensembles of closely packed nanocrystals (NCs). We study NCs in a diluted regime. The density of NCs is estimated to be about $10^{15}/\text{cm}^3$, see [Kolobkova] and Supporting information in [Nestoklon]. Recent studies have demonstrated large differences in the coherent dynamics of closely packed ensembles and single NCs [Raino, Blach, Levy]. It is possible that closely packed ensemble has a different energy structure due to interactions between the NCs which influence the coherent dynamics after excitation with fs pulses. The differences between these two cases require further studies. We note that our case is closer to the ideal situation of non-interacting NCs, as also described in our model.

In addition, there are material constraints, that also make the studies distinctly different. In order to establish the fully coherent regime and to observe the quantum beats between exciton-polarons the following features must be satisfied: Firstly, low temperatures are required to ensure that the zero-phonon T₂ time is sufficiently long. Secondly, narrow phonon modes are preferable. Thirdly, the exciton-phonon interaction should be strong enough to enable coupling but not excessively large.

In our study, we examine CsPbI₃ perovskite nanocrystals, whereas Ref. [35] investigates CsPbBr₂Cl perovskite nanocrystals. There are distinct material differences between the two systems: in bromides, the optical phonon modes are relatively broad [Cohen, Iaru], which leads to rapid dephasing of excited exciton-polaron states such as O' and X'. The exciton-phonon interaction strength is also different in bromide NC, hence the optimal conditions were possibly not met in Ref.[35].

In Ref. [44] the authors focused on features related to the triplet state fine structure with beating frequencies below 4 meV and deferred from a discussion of vibrational features (as stated on page 2 of Ref. [44]).

Changes in the text:

In the Introduction, we emphasize that most previous coherent spectroscopy studies have been performed on closely packed NC ensembles, as noted in the penultimate paragraph (page 2).

In Section IV we emphasize that observation of pronounced quantum beats requires optimal choice of parameters, i.e. long T_2 , well resolved phonon frequencies and optimal HR factors. End of 2nd, beginning of 3rd paragraph (page 6).

References:

[Kolobkova] E. V. Kolobkova, M. S. Kuznetsova, and N. V. Nikonorov, Perovskite CsPbX₃ (X=Cl, Br, I) Nanocrystals in fluorophosphate glasses, *Journal of Non-Crystalline Solids* **563**, 120811 (2021). <https://doi.org/10.1016/j.jnoncrysol.2021.120811>

[Nestoklon] M. O. Nestoklon, E. Kirstein, D. R. Yakovlev, E. A. Zhukov, M. M. Glazov, M. A. Semina, E. L. Ivchenko, E. V. Kolobkova, M. S. Kuznetsova, and M. Bayer, Tailoring the electron and hole Landé factors in lead halide perovskite nanocrystals by quantum confinement and halide exchange, *Nano Letters* **23**, 8218 (2023). <https://doi.org/10.1021/acs.nanolett.3c02349>

[Raino] G. Rainò, M. A. Becker, M.I. Bodnarchuk, Superfluorescence from lead halide perovskite quantum dot superlattices. *Nature* **563**, 671 (2018). DOI: 10.1038/s41586-018-0683-0

[Blach] D.D. Blach, V.A. Lumsargis, D.E. Clark, Superradiance and Exciton Delocalization in Perovskite Quantum Dot Superlattices. *Nano Lett.* **2022**, *22*, 7811-7818, DOI: 10.1021/acs.nanolett.2c02427

[Levy] S. Levy, O. Be'er, S. Shaek, A. Gorlach, E. Scharf, Y. Ossia, R. Liran, K. Cohen, R. Strassberg, I. Kaminer, U. Banin, and Y. Bekenstein, Collective Interactions of Quantum-Confined Excitons in Halide Perovskite Nanocrystal Superlattices, *ACS Nano* **2025** *19* (1), 963-971 DOI: 10.1021/acsnano.4c12509

[Cohen] A. Cohen, T. M. Brenner, J. Klarbring, R. Sharma, D. H. Fabini, R. Korobko, P. K. Nayak, O. Hellman, O. Yaffe, Diverging expressions of anharmonicity in halide perovskites. *Adv. Mater.* **2**, 2463–2469 (2017). <https://doi.org/10.1002/adma.202107932>

[Iaru] C. M. Iaru, A. Brodu, N. J. J. van Hoof, S. E. T. ter Huurne, J. Buhot, F. Montanarella, S. Buhbut, P. C. M. Christianen, D. Vanmaekelbergh, C. de Mello Donega, J. G. Rivas, P. M. Koenraad, and A. Y. Silov, Fröhlich interaction dominated by a single phonon mode in CsPbBr₃, *Nature Communications* **12**, 5844 (2021). <https://doi.org/10.1038/s41467-021-26192-0>

Q1.2. Related to the comment above, there indeed exists a previous study in which the exciton fine-structure coherence (not equivalent to exciton coherence, though) coexist with phonon coherence. In Nat. Mater. 2022, 21, 1282–1289, Han reported simultaneous observation of exciton fine-structure coherence and phonon coherence. The phonon modulation is very shallow, but it can be observed at high pump fluences (Supplementary Fig 4 therein) and the phonon beating frequency (about is also much faster than the fine-structure quantum beating. The authors should discuss the distinction of the present study from that work.

R1.2. We thank the referee for pointing out the interesting observation of coherent optical phonons under resonant excitation of excitons in CsPbI₃ nanocrystals.

Hahn et al. reported the simultaneous observation of exciton fine-structure coherence and phonon coherence at low temperature. The latter indicates inter-polaron coherence, i.e., a coherent superposition between polaron states X and X' with different energies, which is a prerequisite for entering the fully coherent regime. Our results, namely the observation of quantum beats in emission, directly demonstrate this fully coherent regime, in which optical coherence (i.e., superpositions between the ground states 0 and 0' and the exciton-polaron states X and X') is preserved and is temporally limited by the lifetime of the excited polaron state X' or the phonon state 0'.

We note that transient absorption spectroscopy does not necessarily probe the population of the same states that are created by the pump pulse, because relaxation processes may occur between excitation and detection (see also responses R2 and R2.2 and references [Thouin, Debnath, Han]).

Changes in the text.

We explained the differences between fully coherent dynamics of exciton-polaron states in the coherent optical response and the coherent inter-polaron dynamics in pump-probe studies in the introduction section (3rd paragraph, pages 1-2). We also included the references below.

We clarified the states notation in the model (changes in the first paragraph of section IV, page 5) and the importance of optical coherence in the fully coherent regime for observation of quantum beats in the emission (end of 2nd paragraph in section IV, page 6).

References:

[Thouin] F. Thouin, D. A. Valverde-Chávez, C. Quarti, D. Cortecchia, I. Bargigia, D. Beljonne, A. Petrozza, C. Silva, and A. R. Srimath Kandada, Phonon coherences reveal the polaronic character of excitons in two-dimensional lead halide perovskites, *Nature Materials* 18, 349 (2019). <https://doi.org/10.1038/s41563-018-0262-7>

[Debnath] T. Debnath, D. Sarker, H. Huang, Z.-K. Han, A. Dey, L. Polavarapu, S. V. Levchenko, and J. Feldmann, Coherent vibrational dynamics reveals lattice anharmonicity in organic-inorganic halide perovskite nanocrystals, *Nature Communications* 12, 2629 (2021). <https://doi.org/10.1038/s41467-021-22934-2>

[Han] Y. Han, W. Liang, X. Lin, Y. Li, F. Sun, F. Zhang, P. C. Sercel, and K. Wu, Lattice distortion inducing exciton splitting and coherent quantum beating in CsPbI₃ perovskite quantum dots, *Nature Materials* 21, 1282 (2022). <https://doi.org/10.1038/s41563-022-01349-4>

Q1.3. The sensitivity of the exciton T₂ to the nanocrystal size seems to be too sensitive. The authors basically select different exciton energies from the ensemble by tuning the laser frequencies. But the 100 fs pulse itself is very broad, so the energy selection is not that accurate. In this case, it is hard to imagine that in such a small energy range (1.724 to 1.75 eV), T₂ varies from 330 to 230 ps. How accurate are the fitting parameters? Error bars should be given.

R1.3 We thank the referee for the question. The reviewer expressed concerns about whether the spectral dependence of T₂ can be reliably resolved when using femtosecond (fs) pulses, due to their limited spectral resolution (spectral width of 15 meV). We note that due to its nonlinear origin, the photon echo exhibits a spectrally narrower response than the laser bandwidth, with a width of about 10 meV estimated from the echo transient profile.

The figure above shows the photon energy dependence of T_2 measured using picosecond (ps) and femtosecond (fs) laser pulses. The use of picosecond pulses (0.5 meV spectral width) removes concerns related to the limited spectral resolution of fs pulses. For excitation with fs pulses similar values are obtained (see red symbols in the Figure). This is in accordance with our expectations, because the spectral dependence is smooth and varies only weakly on the scale of 10 meV. Therefore, we consider that our experiments with fs pulses address appropriately the dependence of HR factors and phonon lifetimes in Fig. 4 for NCs with different sizes. We note that the studied spectral range is relatively small, corresponding to a change in the quantum confinement energy from 73 meV to 113 meV only [NL 23, 8218 (2023); NL 25, 12754 (2025)]. Nevertheless, the latter corresponds to an approximately twofold change in the effective NC volume, leading to noticeable changes in T_2 , τ_{ph} , and S_{HR} .

Regarding the dependence of T_2 on excitation energy, it should be noted that the measured T_2 is smaller than the lifetime limit $2T_1$. The exciton lifetime in these NCs is given $T_1=500$ ps as follows from time-resolved PL data in Ref. [PRB 113, 035304 (2026)]. This indicates that the observed coherence time T_2 is not limited by lifetime broadening but is instead governed by phase relaxation processes (pure decoherence). The strong temperature sensitivity, i.e. a decrease of T_2 from ~ 300 ps to ~ 50 ps upon heating from 2 K to 5 K, clearly points to phonon-related mechanisms. It is therefore reasonable to assume that the observed spectral dependence of T_2 is determined by the specifics of exciton–phonon coupling, most likely involving acoustic phonons with energies of 0.5–1 meV at such low temperatures. One possible scenario is that changes in the nanocrystal size modify the degree of resonant interaction between acoustic phonons and the fine-structure split excitons. This scenario is consistent with the work mentioned by the reviewer (Han et al., Nat. Mater. 2022, 21, 1282–1289).

Changes in the text:

Spectral dependence and discussion are shown in the Supplementary Information Sec.S2 and Figure S1.

We specify that the data for long-lived coherence in Fig. 1(d) are measured with ps excitation pulses (description of panel (d) in the figure caption)

We added a description of the picosecond laser pulses used for the measurements in Fig. 1(d) to the Methods section (page 8).

Added horizontal error bars in Fig.4 corresponding to the spectral width of PE signal +/- 5 meV

Q1.4. The authors may consider the anharmonicity effect in the observed the phonon dephasing dynamics. The method is well documented in the literature (e.g., J. Phys. Chem. Lett. 2022, 13, 5413–5423). It would be interesting to compare the anharmonicity of the two optical phonon modes at 5.1 and 3.2 meV, as well as the effect of nanocrystal size on this anharmonicity.

R1.4 We tried to analyze the anharmonicity effect by assuming a different energy of optical phonons in the ground crystal and exciton-polaron states. The associated changes of dynamics are not observed in the experimental data. Furthermore, since experimental data do not cover two- and three- phonon states, we have no direct evidence of an anharmonicity effect and correspondingly cannot measure it. We believe that for the phonon dynamics the interaction with acoustic modes is more important. The anharmonicity is demonstrated to be important starting from ~80K, see [Ferreira].

We also note that we observe second-harmonic components, which should not be confused with anharmonic effects. These components arise from the interference between the $0'-X$ and $0-X'$ transitions, which differ in energy by $2\hbar\Omega$. Such interference is possible only in a two-pulse experiment and represents a genuine feature of the detected nonlinear four-wave mixing signal.

Changes in the text:

We included additional discussion in the beginning of 3rd paragraph of Section IV (page 6).

Added analytic expressions (S54, S55) and discussion for Fourier spectrum in the end of Supplementary Section S4.C.

References:

[Ferreira] A. C. Ferreira, S. Paofai, A. Létoublon, J. Ollivier, S. Raymond, B. Hehlen, B. Rufflé, S. Cordier, C. Katan, J. Even, and P. Bourges, Direct evidence of weakly dispersed and strongly anharmonic optical phonons in hybrid perovskites, *Communications Physics* 3, 1 (2020). <https://doi.org/10.1038/s42005-020-0313-7>

Response to Reviewer #2 (Remarks to the Author):

Rev2: This manuscript reports on exciton-lattice interactions in perovskite QD, specifically CsPbI₃. The topic itself is of paramount importance and is a major question in quantum materials. But there have been many efforts to address this topic using femtosecond and single dot PL spectroscopies. So it must be clear in the paper what is new in 2026. That point is far from obvious in this paper. It is an excellent paper. The data is lovely. Their writing is great. But what is new? Moreover some of the main concepts of lattice dynamics are not correct. Details are below.

R2: We thank the referee for the careful reading of the manuscript and for the valuable questions, which helped us to clarify the main result of our work, in particular with regard to previous works.

The novelty of our work is the observation of **quantum beats from exciton-polaron states** in the fully coherent regime. To our knowledge such kind of dynamics was not observed in halide perovskites earlier. Moreover, exciton-polaron quantum beats have not been demonstrated with such high efficiency and over such long time scales in other solid-state systems.

To reach the fully coherent regime, we use resonant excitation with short laser pulses that operates with a pure quantum state, in which the phases of the coherent superposition of all eigenstates (zero-phonon ground state 0 , excited single-phonon state $0'$, zero-phonon exciton-polaron X , and first excited exciton-polaron X') are well defined. Emission from such a pure quantum state exhibits genuine quantum beats. At the level of a single nanocrystal, this means that it is not possible to determine through which quantum pathway the photon was emitted. This effect can be exploited for the generation of a single optical phonon in a nanocrystal and for the demonstration of photon-phonon entanglement.

We emphasize that our work goes beyond previous pump-probe studies, which have indeed made significant contributions to the understanding of polaron dynamics in lead halide perovskites. These studies addressed the optical generation of coherent phonon wavepackets, exciton population dynamics, and polaron formation, primarily under non-resonant excitation and at elevated temperatures. In contrast, our work approaches exciton-polaron dynamics from a different perspective, namely the fully coherent regime, where resonant optical excitation addresses directly exciton-polaron states dressed by only few phonons in a nanocrystal, i.e., we consider the discrete exciton-polaron spectrum at a quantum level.

Most importantly, the two-pulse experiment probes only the states that are directly excited. In other words, we resonantly excite the exciton-polaron in the ground X and excited X' eigenstates and monitor exclusively their quantum dynamics, because any energy or phase relaxation into other states would lead to the decay of the photon echo. We thus demonstrate that, at low temperatures, quantum beats (i.e., a pure quantum state) persist on a timescale of about 10 ps and are limited by the lifetime of the excited exciton-polaron state. This approach also enables a straightforward evaluation of the Huang-Rhys factors, i.e., the exciton-phonon coupling strength. We emphasize that such a fully coherent regime has not been established in previous studies.

Changes in the text: We revised abstract and introduction (3rd paragraph, pages 1-2) in order to strengthen the novelty of our approach to the investigation of exciton-polaron dynamics.

Q2.1. The abstract and title move between phonons and polarons. Both are relevant and care must be taken to relate the two. Phonons exist at 2K as normal modes that are coherent. Polarons exist at 300

K as local modes from the basis of normal modes, and are diffusive. They are related but not the same. This point must be addressed up front.

R2.1 We thank the referee for this remark. We want to clarify, that polarons can be found in many different systems and the understanding of these quasiparticles and their behavior differs strongly depending on the dimensionality of the system. In our work, we consider electron-hole pairs which are represented by excitons both in bulk semiconductors and in quantum dots, that interact with the phonon modes forming an exciton-polaron. The polaron properties that the referee is referring to are typically found for free polarons in bulk or two-dimensional perovskite semiconductors. These polarons differ strongly in their behavior to the localized polarons in zero-dimensional case discussed in our manuscript.

In our work we demonstrate that the experimental data may be understood both qualitatively and quantitatively within a simplified model which assumes direct optical transitions into the exciton-polaron (exciton "dressed" by the interaction with particular optical phonon modes) states. This simple four-level model gives a good description of the experimental data. We do not claim to have an explanation of the structure of (localized) exciton-polaron at elevated temperatures where our model is obviously not relevant. We also stress that, in our work, the ground-state energy ladder is associated with phonons, while the optically excited states are treated as a ladder of exciton-polaron states.

Nonetheless, we agree that the terminology might be confusing, as we sometimes refer to the exciton-polarons simply as "polarons" or "excitons." In all cases, the correct term is "exciton-polarons," and we have revised the manuscript to use this terminology consistently.

Changes in the text:

We have revised terminology throughout the manuscript, particularly in Sec. IV (page 5) and Supplementary Information Sec. 4.

We added the following sentence in the 2nd paragraph of the introduction section (page 1): "Exciton-polarons are defined here as confined exciton states in the nanocrystal that are instantaneously dressed by interaction with phonons."

Q2.2. The abstract discusses the importance of lattice dynamics but in the end only measures exciton-phonon coupling strength. More should be shown.

R2.2. We measure the coherent emission of a pure quantum state after resonant excitation with a femtosecond (fs) laser pulse and observe pronounced quantum beats between the exciton-polaron states. In this respect, the coherent dynamics of the exciton-polaron system is fully resolved and is well described by a four-level model, which is applicable in the regime of moderate Huang-Rhys factors. Our measurements resolve long-lived zero-phonon coherence (0-X, >200ps) as well as the lifetime of the pure quantum state governed by the phonon lifetimes (i.e., coherent superpositions involving the 0' and X' states). We observe a correlation between the exciton-phonon coupling strength (0.02-0.12) and the decay time of the quantum beats, which we attribute to variations in the phonon lifetime (5-15 ps).

Therefore, we identify the physical mechanism governing the quantum beats between exciton-polaron states in perovskite nanocrystals with moderate Huang-Rhys factors. The decoherence on longer timescales and the dynamics in the strong-coupling regime ($HR > 1$) require further investigation, and our work motivates such studies.

Changes in the text: We have revised the abstract and included numbers highlighting the dynamics.

Q2.3. The authors review the literature on phonons in perovskite QD. Some are cited here(1-10). So what is new?

R2.3 We thank the referee for providing relevant references. Some of them were already included in the original manuscript, and additional relevant references have been incorporated into the revised version. Of particular interest are studies employing pump-probe transient absorption techniques, which demonstrated optical excitation of vibrational wave packets, polaron formation, and carrier population dynamics modulated by coherent and thermal phonons [1-5]. However, none of these studies addressed the fully coherent regime and therefore could not observe quantum beats between polaron states in emission (see also our response R2).

Despite the rich and comprehensive population dynamics revealed by pump-probe studies, this approach has several limitations:

1. Pump-probe techniques provide access to population dynamics and inter-excitonic coherences but do not allow direct measurement of optical coherence.
2. Transient absorption does not necessarily probe the population of the same states that are created by the pump pulse, because relaxation processes may occur between excitation and detection (see also response R1.2). We note that Refs. [1-3,5] employed either off-resonant excitation (i.e., excitation and probing of different states) or room-temperature conditions, under which a fully quantum-mechanical description of polaron states is not required.
3. While pump-probe spectroscopy allows evaluation of Huang-Rhys factors, photon-echo techniques provide a clean separation of homogeneous broadening from inhomogeneous contributions.

In Ref. [6] based on Raman spectra and DFT calculations the authors conclude that the anharmonicity is important for the central peak in the Raman signal. Without clear quantification of this anharmonicity, we may only speculate that anharmonic modes are important for the results of high-temperature Raman spectra and do not affect the low-temperature photon echo signal. However, they may be responsible for the fast decay of T_2 as a function of temperature, which is out of the scope of present investigation.

References [7–11] based on photoluminescence spectroscopy addressed the strength of exciton-phonon coupling via phonon replicas in the emission spectra. However, these works did not report quantum beats between polaron states and did not address the dynamical processes or coherence times associated with excited exciton-polaron states.

Therefore, our study establishes a new approach to the exciton-polaron dynamics in perovskite NCs with an explicit focus on optical coherence and its temporal evolution in the form of quantum beats.

Changes in the text:

We revised the introduction and added relevant references [1,4,5,9] from the reviewer list (pages 1 and 2). Other refs. [7,8,10,11] were already cited.

Q2.4. The authors discuss the long lived electronic coherence of the exciton. But is this really long-lived coherence? IN 2DE and CMDS, long-lived coherence means the electronic coherence between excitonic states is long lived relative to some internal clock such as the T_2 of the basis excitons. Such as case has been found for inter exciton coherence at 300 K (5).

R2.4 We thank the referee for pointing out this interesting reference. The long lived coherence reported in Ref. [5] is not an optical coherence but an inter-exciton coherence. In contrast, our work focuses exclusively on optical coherences and on quantum beats in a multi-level system, rather than on inter-exciton or inter-polaron coherences. A well-established example illustrating this distinction is the fine-structure spin coherence, whose coherence time can significantly exceed that of the optical coherence. In this context, optical coherence can be also transferred to much longer-lived spin coherences [Nature Photonics 8, 851 (2014); PRB 103, 235312 (2021)]. However, such coherence transfer mechanisms are beyond the scope of the present work.

In the context of optical coherence, a coherence time T_2 comparable to the population decay time T_1 can indeed be considered long-lived. Therefore, we refer to the zero-phonon optical coherence (0-X) as long-lived, and note that it exhibits a record value among currently reported lead halide perovskite nanocrystals. We emphasize that optical coherences involving excited exciton-polaron states (e.g., 0-X') or inter-polaron coherences (X-X') are not long-lived in the studied system.

Changes in the text: We included the reference [5] in the introduction section (page 2) on the current state of the art (paragraph 3).

Q2.5. The synthetic methods of the QD are not the state of the art as shown by other authors such as Kovalenko or DH Son. As a result great care must be taken in the characterization of them. There must be much information about them in the SI.

R2.5: We thank the referee for this comment. The characterization of these samples has been discussed in previous papers [NL 23, 8218 (2023), Nanoscale 16, 21496 (2024), Nanoscale 17, 6522 (2025), NL 25, 12754 (2025), PRB 113, 035304 (2026)]. We included the corresponding summary of their properties in the Supplementary section.

Changes in the text: We have added Supplementary Section 1 with the main properties of the studied NCs.

Q2.6. What is the 2K T_2 compared to single dot experiments by Bawendi (11)?

R2.6: The zero-phonon optical coherence time T_2 in our measurements corresponds to 330 ps which is about 6 times longer compared to the reference mentioned by the Reviewer (52 – 78 ps). We emphasize that the reported value in our study is a record value due to the low temperature of $T=2K$. At larger temperatures scattering on acoustic phonons becomes important.

Changes in the text: We included a table with coherence times measured by different techniques in halide perovskite NCs (Supplementary section 2).

Q2.7. The most interesting FFT signal is the zero frequency rolloff. It is not seen in Raman because CW methods cannot measure low frequency accurately but time domain can. So what is this roll off? Is that the residual thermal disorder of pre polarons at 2K?

R2.7: The behavior of the photon echo FFT near zero frequency reflects the slow temporal dynamics of the system. In our work we use an original technique to disentangle the signal originating from the optical phonons and the signal originating from the fine structure. The value Σ (shown in Fig. 2c) which

we associate with phonons, at low temperatures demonstrates mostly Lorentzian shapes, as expected for the exponential decay of the signal.

Most importantly, according to our model the real part of the Fourier transform of the echo signal is dominated by the sum of a few Lorentzians with different widths. In addition to the main narrow Lorentzian with the half width at half maximum (HWHM) of $\approx\gamma_0$ at zero frequency there are a few contributions with the amplitudes proportional to the Huang-Rhys factor. These contributions have larger broadening which is determined by γ_{ph} (for each phonon mode): (i) at zero frequency; (ii) at the phonon frequency Ω ; and (iii) at the double phonon frequency 2Ω . The amplitudes of the zero- and second-harmonic Lorentzians, with HWHM $\approx\gamma_{ph}$ are approximately 4 times smaller than the amplitude of the peak at the fundamental phonon frequency. Therefore, at frequencies around zero the roll off contains several contributions with narrow zero phonon linewidth $\approx\gamma_0$ and broader phonon assisted peaks $\approx\gamma_{ph}$. At low temperatures this is in full accord with the experimental data shown in Fig.4 and discussed in the text in Sec. IV.

Here, we emphasize that the long-lived signal decay is governed by T_2 , which is not limited by the exciton lifetime ($T_1 = 500$ ps) but by additional pure dephasing. The strong temperature dependence, i.e., the decrease of T_2 from ~ 300 ps to ~ 50 ps upon increasing the temperature from 2 K to 5 K, clearly indicates phonon-related mechanisms. Therefore, we assume that exciton-phonon coupling is responsible for pure dephasing, most likely involving acoustic phonons [Harkort] with energies of 0.5-1 meV at such low temperatures. Note that this range of energies is comparable to the fine structure splitting in these NCs. The part of the signal demonstrating pronounced polarization dependence (ρ in Fig. 2c) indeed demonstrates a very intricate behavior around zero frequency. We associate this signal with the fine structure of the exciton. A more detailed analysis involving the details of the fine structure and its interplay with acoustic phonons requires further studies.

Changes in the text:

We have added Eqs. S54 and S55 describing the FFT spectra, along with a discussion of the signal containing several Lorentzians at zero frequency (roll-off) in Supplementary Information Sec. 4.

Spectral dependence of T_2 indicating importance of pure decoherence on acoustical phonons and a possible link with the fine structure is discussed in Supplementary Information Sec. 2.

References:

[Harkort] C. Harkort, I. V. Kalitukha, N. E. Kopteva, M. O. Nestoklon, S. V. Goupalov, L. Saviot, D. Kudlacik, D. R. Yakovlev, E. V. Kolobkova, M. S. Kuznetsova, and M. Bayer, Confined acoustic phonons in CsPbI₃ nanocrystals explored by resonant Raman scattering on excitons, Nano Lett. 25, 12754 (2025). <https://doi.org/10.1021/acs.nanolett.5c03342>

Q2.8. Why is the coupling, S , closer to 0.1 instead of 1 as seen in other measurements?

R2.8. We are unaware of experimental data on a Huang-Rhys factor close to 1 in CsPbI₃ nanocrystals. To our knowledge, a Huang-Rhys factor in the range 0.1-0.6 (with the larger values corresponding to very small NCs) has been estimated in CsPbBr₃ and FAPbBr₃ nanocrystals [Ref (7) in the reviewer report] based on single NC photoluminescence. In analogy with data on bulk materials, we expect that the electron-phonon interaction in iodide perovskites is weaker than in bromide ones. A value of the Huang-Rhys factor close to 1 has been estimated for Dion–Jacobson-type two-dimensional perovskite (FPT)PbI₄ in [Ref. (1) in report]. However, we do not think that this value can be directly transferred to

our system. Note that in the same work a significantly smaller value is reported for D-J material with a different organic component.

References:

References in the Referee report:

1. S. Biswas et al., Exciton polaron formation and hot-carrier relaxation in rigid Dion–Jacobson-type two-dimensional perovskites. *Nature Materials*, 1-7 (2024).
2. X. Yue et al., Real-time observation of the buildup of polaron in α -FAPbI₃. *Nat. Commun.* 14, 917 (2023).
3. C. D. Sonnichsen, D. P. Strandell, P. J. Brosseau, P. Kambhampati, Polaronic quantum confinement in bulk CsPbBr₃ perovskite crystals revealed by state-resolved pump/probe spectroscopy. *Physical Review Research* 3, 023147 (2021).
4. F. Thouin et al., Phonon Coherences Reveal the Polaronic Character of Excitons in Two-Dimensional Lead Halide Perovskites. *Nat. Mater.* 18, 349 (2019).
5. A. Ghosh et al., Correlated Lattice Fluctuations in CsPbBr₃ Quantum Dots Give Rise to Long-Lived Electronic Coherence. *ACS Nano* 19, 19927-19937 (2025).
6. O. Yaffe et al., Local Polar Fluctuations in Lead Halide Perovskite Crystals. *Phys. Rev. Lett.* 118, 136001 (2017).
7. C. Zhu et al., Quantifying the size-dependent exciton-phonon coupling strength in single lead-halide perovskite quantum dots. *Advanced Optical Materials* 12, 2301534 (2024).
8. K. Cho et al., Exciton–phonon and trion–phonon couplings revealed by photoluminescence spectroscopy of single CsPbBr₃ perovskite nanocrystals. *Nano Lett.* 22, 7674-7681 (2022).
9. O. Pfingsten et al., Phonon interaction and phase transition in single formamidinium lead bromide quantum dots. *Nano Lett.* 18, 4440-4446 (2018).
10. M. Fu et al., Unraveling exciton–phonon coupling in individual FAPbI₃ nanocrystals emitting near-infrared single photons. *Nat. Commun.* 9, 3318 (2018).
11. H. Utzat et al., Coherent Single-Photon Emission from Colloidal Lead Halide Perovskite Quantum Dots. *Science* 363, 1068 (2019).

We thank the referee for pointing to several references. The relevant references 1,4,5 on pump-probe and 9 on PL of single NCs are included. The other references (7,8,10,11) were cited in the original manuscript.

Response to Reviewer #3 (Remarks to the Author):

This manuscript reports the observation of exceptionally long coherence times of exciton-polarons in CsPbI₃ nanocrystals, which is interesting and novel in quantum physics. The underlying mechanism is analyzed through a profound size dependent theoretical calculation. This study can be accepted if the following questions are addressed:

R3. We thank the reviewer for the stimulating questions and constructive comments, which have helped us to improve the manuscript. We address all points below.

Q3.1 In Fig.1a. one may observe multi-peaks configuration, what's the reason of occurrence of these peaks? If they origin from nanocrystals with different sizes, considering the size will affect the decay times of the phonons, a detailed analysis or supporting data should be made to clarify the size effect.

R3.1 The NC ensemble in our samples does not exhibit a Gaussian (normal) size distribution, which is commonly assumed for colloidal NCs. In the present work, we study NCs grown in a glass matrix using a rapid melt-quenching technique. The growth mechanism in this case can naturally lead to an unconventional size distribution.

In an oversimplified picture, the NCs form from the precursor materials dissolved in the molten glass, while the actual nucleation and growth occur during the rapid cooling of the melt. The final NC size is determined by the cooling rate. During the fabrication, the molten glass is cast onto a cold (room-temperature) glass-ceramic surface. Considering the inhomogeneous cooling rate across the thickness of the sample, as well as the nonlinear dependence of the NC size on the cooling rate, one can expect a non-Gaussian distribution of NC sizes (and possibly even of crystalline phases).

There is a number of factors which might lead to the observed multi-peak configuration:

1. All luminescent phases of perovskite nanocrystals are metastable. The formation of the γ (orthorhombic) and α (cubic) phases have comparable probability of stabilization in the nanocrystalline state, as has been demonstrated for colloidal nanocrystals. The bandgap energies of these phases differ, which can result in two distinct nanocrystal ensembles contributing to the luminescence.
2. The second emission band can be associated with the presence of shallow trap states which do not contribute to the coherent spectroscopy data.
3. A purely technological factor: The cooling rate of the surface layers is significantly higher than that of the bulk and the size distribution in the sample volume is not purely homogeneous.

However, we stress that the resonant excitation in the photon echo technique allows us to study only a sub-ensemble of optically active nanocrystals with distinct size. Different spectroscopic techniques do not always probe the same states as photoluminescence (PL) and can therefore show a significantly altered spectral response, especially for signals arising from excitonic transitions in perovskites. For instance, in Raman spectroscopy the spectral range where Raman signal may be detected is much narrower than the full PL width, see Ref. [Harkort], Figure 1a-d.

Changes in the text: We have introduced description of the optical properties in Supplementary Section S1 where inhomogeneities are discussed (see also R2.5).

References:

[Harkort] C. Harkort, I. V. Kalitukha, N. E. Kopteva, M. O. Nestoklon, S. V. Goupalov, L. Saviot, D. Kudlacik, D. R. Yakovlev, E. V. Kolobkova, M. S. Kuznetsova, and M. Bayer, Confined acoustic phonons

in CsPbI₃ nanocrystals explored by resonant Raman scattering on excitons, Nano Lett. 25, 12754 (2025). <https://doi.org/10.1021/acs.nanolett.5c03342>

Q3.2. For nanocrystals with different sizes, the same phonon energies $\hbar\Omega_1 = 3.2$ meV and $\hbar\Omega_2 = 5.1$ meV are used however, this is not necessarily the case, the phonon energies of different nanocrystals should be investigated.

R3.2. The optical phonon dispersion might be important for small nanocrystals. Though, this dispersion is relatively flat the energy can differ at the boundary of the Brillouin zone from the energy at the Γ point in the bulk case. In the analysis of the PE signal, we used the phonon energies as fitting parameters and found them constant within the accuracy of experiment (see Figure below). The error bars correspond to the width of the Fourier peaks given by $\pm\hbar/\tau_{\text{ph}}$.

Changes in the text: We included discussion of the size dependence of the phonon frequencies in Supplementary Information Sec. 5 and corresponding Figure S2.

Q3.3. In the main text the author wrote "making CsPbI₃ NCs embedded in a glass matrix particularly interesting for investigating coupled exciton-phonon dynamics." But they did not clarify how the glass matrix impacts the exciton-phonon coupling? Is this glass matrix crucial for the observation of the long coherent times of exciton polaron?

R3.3. The importance of the glass matrix is not yet fully understood. There are a few possible reasons why these NCs demonstrate long coherent times, as compared with other samples. The mechanisms of decoherence critically depend on acoustic phonons, which are well confined in the glass samples [Harkort], as compared with arrays of colloidal nanocrystals [Qian]. In addition, the collective optical effects (e.g. superfluorescence, see Ref. [Raino]) are suppressed in a diluted glass solution (see also R1.1).

Changes in the text: we included explanation in the introduction about closely packed NCs (page 2).

References:

[Harkort] C. Harkort, I. V. Kalitukha, N. E. Kopteva, M. O. Nestoklon, S. V. Goupalov, L. Saviot, D. Kudlacik, D. R. Yakovlev, E. V. Kolobkova, M. S. Kuznetsova, and M. Bayer, Confined acoustic phonons in CsPbI₃ nanocrystals explored by resonant Raman scattering on excitons, *Nano Lett.* 25, 12754 (2025). <https://doi.org/10.1021/acs.nanolett.5c03342>

[Qian] C. Qian, E. Stanifer, Z. Ma, L. Yao, B. Luo, C. Liu, J. Li, P. Pan, W. Pan, X. Mao, and Q. Chen, Nanoscale phonon dynamics in self-assembled nanoparticle lattices, *Nature Materials* 24, 1616 (2025). <https://doi.org/10.1038/s41563-025-02253-3>

[Raino] G. Rainò, M. A. Becker, M. I. Bodnarchuk, R. F. Mahrt, M. V. Kovalenko, and T. Stöferle, Superfluorescence from lead halide perovskite quantum dot superlattices, *Nature* 563, 671 (2018). <https://doi.org/10.1038/s41586-018-0683-0>

Q3.4. The exciton coherent time is dependent on the photon energy, why?

R3.4. Indeed, the coherence time T_2 depends on the photon energy, which can be attributed to its dependence on the NC size. The optical coherence time T_2 is smaller than the exciton lifetime limit of $2T_1$. The exciton lifetime in these NCs is given by $T_1=500$ ps as follows from the time-resolved PL data in [Meliakov2026]. This indicates that the observed coherence time T_2 is not limited by lifetime broadening but is instead governed by phase relaxation processes (pure decoherence). The strong temperature sensitivity, i.e. the decrease of T_2 from ~ 300 ps to ~ 50 ps upon heating from 2 K to 5 K, clearly points to phonon-related mechanisms. It is therefore reasonable to assume that the observed spectral dependence of T_2 is determined by the specifics of the exciton-phonon coupling, most likely involving acoustic phonons with energies of 0.5–1 meV at such low temperatures. One possible scenario is that changes in the nanocrystal size modify the degree of resonant interaction between acoustic phonons and the fine-structure split excitons, consistent with [Han]. (see also response R1.3).

Changes in the text: We have added Supplementary Information Sec.2 where the spectral dependence of T_2 is discussed.

References:

[Meliakov] S. R. Meliakov, E. A. Zhukov, V. V. Belykh, K. V. Kavokin, M. O. Nestoklon, E. V. Kulebyakina, M. L. Skorikov, E. V. Kolobkova, M. S. Kuznetsova, M. Bayer, and D. R. Yakovlev, Hyperfine interaction of electrons confined in cspbi₃ nanocrystals with nuclear spin fluctuations, *Phys. Rev. B* 113, 035304 (2026). <https://doi.org/10.1103/s33c-m6hz>

[Han] Y. Han, W. Liang, X. Lin, Y. Li, F. Sun, F. Zhang, P. C. Sercel, and K. Wu, Lattice distortion inducing exciton splitting and coherent quantum beating in CsPbI₃ perovskite quantum dots, *Nature Materials* 21, 1282 (2022). <https://doi.org/10.1038/s41563-022-01349-4>

Q3.5. In ref[15][Nat. Mater. 18, 717–724 (2019)], the bright-dark exciton splitting would be determined by the crystal local symmetry and quantum confinement, I suggest the consideration of this effect for the discussion of the exciton-polarons.

R3.5. In our experiment in zero magnetic field the dark states are not addressed optically. At low temperatures relaxation into the dark exciton-polaron states is negligible and therefore they do not contribute to the coherent dynamics studied here.

The population relaxation time T_1 , was evaluated from three-pulse photon echo experiments, and found to be $T_1=600 - 800$ ps, which is comparable to the exciton lifetime (500 ps) measured by time-resolved PL on the same sample as also stated in the text in the last paragraph of Section 2.

Changes in the text:

We have included an additional sentence regarding dark states in the first paragraph of section 3 (page 4).

We also added the reference [Tamarat]

References:

[Tamarat] P. Tamarat, M. I. Bodnarchuk, J.-B. Trebbia, R. Erni, M.V. Kovalenko, J. Even and B. Lounis, The ground exciton state of formamidinium lead bromide perovskite nanocrystals is a singlet dark state, Nat. Mater. 18, 717–724 (2019). <https://doi.org/10.1038/s41563-019-0364-x>

Point-by-point response to the reviewers' comments

We would like to thank all referees for the careful reading of the manuscript and their valuable suggestions aimed at improving its clarity and impact.

Response to Reviewer #2 (Remarks to the Author):

I have now read the revised manuscript and the authors' response carefully. In my view, the paper is now in publishable shape and can reasonably be accepted after minor revision.

Overall, I find this to be an interesting and worthwhile contribution. The central experimental result is real and significant: the authors report unusually long zero-phonon optical coherence in glass-embedded CsPbI₃ nanocrystals at cryogenic temperature, together with resolved quantum beats arising from low-energy exciton-phonon structure. In my opinion, this is the true strength of the work. The paper is valuable not simply because oscillations are observed, but because the authors appear to access a coherent-optical regime in which the optical coherence survives long enough that the coupled exciton-phonon manifold becomes directly visible in the photon-echo response. That is a nontrivial result and will be of interest to researchers working in coherent spectroscopy, perovskite nanocrystals, and exciton-phonon coupling.

I also think the revised manuscript is improved in important ways. The authors have answered most of the substantive technical questions reasonably well, and the paper is now more precise about what is, and is not, being claimed. In particular, it is helpful that the authors distinguish the unusually long zero-phonon optical coherence observed in their experiment from the broader notion of long-lived inter-state coherence often discussed in the multidimensional-spectroscopy literature. That clarification makes the paper more accurate and considerably stronger.

Reply: We are pleased to receive a positive evaluation from the reviewer.

My remaining concern is not about the quality of the experiment, but about terminology and conceptual scope. I still think one should be careful with the term polaron at 2 K. In the stronger sense usually associated with dynamical lattice reorganization or real-time polaron formation, I would be cautious about over-interpreting the present data. To my mind, the safer language is that the authors observe phonon-dressed excitonic states in a regime of unusually long optical coherence. That said, I do not regard this as a reason to delay publication. Rather, I would suggest a minor further tightening of language so that the manuscript consistently makes clear that its use of "exciton-polaron" refers to a dressed confined excitonic state, not necessarily to the stronger claim of finite-temperature dynamical polaron birth.

Reply: We thank the Reviewer for the constructive feedback regarding the terminology. Historically the polaron at 2 K is the exciton-polaron with a discrete energy spectrum treated as exciton dressed and renormalized by a lattice distortion [Iadonisi, Matsuura]. This is in contrast to free polarons which are represented by a continuous spectrum and energy relaxation accompanied by the polaron formation dynamics [Even] (section 3). The latter is fundamentally different to the coherent dynamics of exciton-polarons studied in our work. We make the difference clear in the revised version.

1. Extended the definition of exciton polaron and included the corresponding reference (page 1): "Exciton polarons are defined here as confined excitons dressed and renormalized by lattice distortion. This definition is particularly appropriate for quantum dots, where electronic excitations are confined in all three dimensions and the difference between localized and delocalized states is not as drastic as in quasi-1D, -2D, or bulk crystals [Even]."

2. Added references [Iadonisi,Matsuura] in the paragraph after Eq.4.
3. Revised all statements and replaced “polaron” by “exciton-polaron” in order to avoid confusion with free polarons.

[Iadonisi] Iadonisi, G., Bassani, F. Excitonic polaron states and optical transitions. *Il Nuovo Cimento D* 2, 1541–1560 (1983). <https://doi.org/10.1007/BF02460231>

[Matsuura] M. Matsuura and H. Büttner, Optical properties of excitons in polar semiconductors: Energies, oscillator strengths, and phonon side bands, *Phys. Rev. B* 21, 679 (1980). <https://doi.org/10.1103/PhysRevB.21.679>

[Even] Jacky Even, Simon Thebaud, Aseem Rajan Kshirsagar, Zeli Xu, Laurent Pedesseau, Marios Zacharias, Claudine Katan, Theoretical approaches to Fröhlich excitonic polarons in polar semiconductors, *arXiv:2505.07406* (2025). <https://doi.org/10.48550/arXiv.2505.07406>

In addition, I think the manuscript would benefit from somewhat broader engagement with the recent coherent multidimensional spectroscopy literature on perovskite quantum dots, especially work bearing on homogeneous lineshape dynamics, correlated lattice fluctuations, coherence, and polaronic interpretation in PQDs. The present paper will sit more naturally and more convincingly within the field if it acknowledges more fully that evolving body of work. In particular, I believe the authors should consider citing the following relevant studies:

*-Seiler, H. et al. Two-dimensional electronic spectroscopy reveals liquid-like lineshape dynamics in CsPbI₃ perovskite nanocrystals. *Nature Communications* 2019, 10, 4962.*

*-Ghosh, A. et al. Correlated lattice fluctuations in CsPbBr₃ quantum dots give rise to long-lived electronic coherence. *ACS Nano* 2025, 19, 19927–19937.*

*-Ghosh, A. et al. Coherent multidimensional spectroscopy reveals homogeneous lineshape dynamics in CsPbBr₃ quantum dots. *ACS Nano* 2025, 19, 26843–26851.*

*-Nagpal, P.; Kambhampati, P. Real-Time Formation of a Landau Polaron. *arXiv* 2026, 2602.24113.*

Reply: The references mentioned by the Reviewer are devoted to studies of free polaron dynamics at room temperature, characterized by a continuous energy spectrum. To clarify this distinction, we have added the following sentence to the Introduction section (page 2):

“It should be noted that two-dimensional Fourier spectroscopy has also been used to resolve free polaron dynamics at room temperature in the adiabatic strong-coupling regime [Seiler, Ghosh1, Ghosh2, Nagpal], a scenario that is fundamentally different from the coherent dynamics of confined exciton-polarons.”

The first two references were included already before. We also added two new references:

[Ghosh2] A. Ghosh, S. Palato, P. Brosseau, R. Tao, D.N. Dirin, M.V. Kovalenko, and P. Kambhampati, Coherent multidimensional spectroscopy reveals homogeneous lineshape dynamics in CsPbBr₃ quantum dots. *ACS Nano* 19, 26843 (2025). <https://doi.org/10.1021/acsnano.5c07429>

[Nagpal] P. Nagpal, A. Ghosh, H. Seiler, S. Palato, and P. Kambhampati, Real-Time Formation of a Landau Polaron, *arXiv* 2026, 2602.24113. <https://doi.org/10.48550/arXiv.2602.24113>

I am not suggesting that the paper be reframed around these works, only that it would be strengthened by recognizing more explicitly the recent CMDS/2DES literature on PQDs and by locating the present result within that broader context.

So my recommendation is straightforward: accept after minor revision. The remaining revisions, in my opinion, should be limited to: ensuring terminological precision around the use of “exciton-polaron” at 2 K, and strengthening the citation context with respect to recent CMDS/2DES work on perovskite quantum dots.

With those minor adjustments, I believe the paper will make a valuable contribution to the literature.

Reply: We made the revision in the terminology and highlighted the difference of our work with respect to the references mentioned by the reviewer.

Response to Reviewer #3 (Remarks to the Author):

The revised manuscript has replied most of my concerns, however, the distribution of the size of the nanocrystals is still lacking, I suggest the authors provide the microscopic images and analysis of the sizes as these are crucial for the explanation of their data.

Reply: The fact that the nanocrystals are embedded in a glass matrix makes a reliable determination of their size and distribution by imaging techniques such as electron microscopy and X-ray diffraction highly challenging, see e.g. the scanning transmission electron microscopy images in Fig.S5 and S6 from Ref. [Nestoklon2023]. Therefore, the size analysis for these samples was performed using optical spectroscopy methods based on the following approaches:

1. Spectral dependence of electron g-factors [Nestoklon2023, Meliakov2025]. It has been demonstrated that for electrons, a pronounced size-induced modification of g-factor occurs due to interaction of the bottom conduction band with the spin-orbit split electron states. This enables a correlation between the spectral dependence of the g-factor and the nanocrystal size.
2. Spectral dependence of Raman peak shifts for light scattering on confined acoustic phonons [Harkort2025]. While optical phonon energies remain largely size-independent, the energies of confined acoustic phonons increase with decreasing NC size. By comparing experimental data with model predictions, it has been demonstrated that the nanocrystals predominantly have spherical or spheroidal shapes.

Based on that studies the photon energy can be recalculated into a NC diameter using the empirical expression $D = \sqrt{16.93/(E_X - 1.652) - 4.31}$ (where E_X is the position of exciton peak in eV) as discussed in the first paragraph of the Sec.: NC size dependence.

We created the new supplementary section S2 (Evaluation of NC sizes) with corresponding explanations.